# Somatostatin expressing GABAergic interneurons in the medial entorhinal cortex preferentially inhibit layer$_{III-V}$ pyramidal cells

Miklós Kecskés [1,2], Nóra Henn-Mike [1,2], Ágnes Agócs-Laboda [1,2], Szilárd Szőcs [1], Zoltán Petykó[1] & Csaba Varga [1✉]

GABA released from heterogeneous types of interneurons acts in a complex spatio-temporal manner on postsynaptic targets in the networks. In addition to GABA, a large fraction of GABAergic cells also express neuromodulator peptides. Somatostatin (SOM) containing interneurons, in particular, have been recognized as key players in several brain circuits, however, the action of SOM and its downstream network effects remain largely unknown. Here, we used optogenetics, electrophysiologic, anatomical and behavioral experiments to reveal that the dendrite-targeting, SOM$^+$ GABAergic interneurons demonstrate a unique layer-specific action in the medial entorhinal cortex (MEC) both in terms of GABAergic and SOM-related properties. We show that GABAergic and somatostatinergic neurotransmission originating from SOM$^+$ local interneurons preferentially inhibit layer$_{III-V}$ pyramidal cells, known to be involved in memory formation. We propose that this dendritic GABA–SOM dual inhibitory network motif within the MEC serves to selectively modulate working-memory formation without affecting the retrieval of already learned spatial navigation tasks.

[1] Szentágothai Research Center, Department of Physiology, Medical School, University of Pécs, Pécs 7624, Hungary. [2]These authors contributed equally: Miklós Kecskés, Nóra Henn-Mike, Ágnes Agócs-Laboda. ✉email: csaba.varga@aok.pte.hu

Specialized GABAergic interneurons control network activity and brain oscillations by innervating different cell types and cellular domains[1]. They release their neurotransmitters in a highly orchestrated manner and provide specific time windows for information processing. The two most abundant and most thoroughly investigated interneuron classes include the perisomatically innervating, parvalbumin (PV)-expressing fast-spiker cells, and the dendrite-targeting, somatostatin (SOM)-expressing interneurons[2–7]. In the hippocampus, SOM$^+$ interneurons regulate the bursting activity of the pyramidal cells, whereas PV$^+$ interneurons control their theta-phase modulation[2]. Importantly, suppressing the activity of either PV$^+$ or SOM$^+$ interneurons does not change nor alter the information conveyed by the firing of place cells in the hippocampal CA1 region[2]. On the other hand, the pharmacological silencing of PV$^+$ interneurons specifically disturbs grid cell activity without affecting the activities of speed cells and border cells in the medial entorhinal cortex (MEC). SOM$^+$ cells, however, do not modulate the activities of the grid, border, and speed cells. Only the information conveyed by nonperiodic spatially active cells is reduced by pharmacogenetic silencing of SOM$^+$ cells[8].

Local network-specific target selectivity of major interneuronal classes in several areas of the temporal cortex has been described. PV$^+$ basket cells, for example, show a strong bias for deep-layer pyramidal cells in the CA1 region, whereas cholecystokinin (CCK) expressing GABAergic basket cells do not show any target selectivity[9]. However, in the MEC, CCK$^+$ interneurons are highly biased to innervate pyramidal cells in layer$_{II}$, and they mostly avoid stellate cells[10]. A major goal of entorhino-hippocampal research is to understand the mechanisms underlying grid-cell firing in the MEC. Given this premise, several models have been proposed[11,12]. Entorhinal cortical-specific network motifs have mostly been investigated in layer$_{II}$[10,11,13,14], where grid cells are the most abundant. However, specific inhibitory actions of local interneurons in deeper layers, where the network is wired for completely different tasks such as coding non-spatial or nonperiodic spatial data and memory formation processes[15,16], have not been revealed.

The classical GABAergic action of the PV$^+$ and SOM$^+$ neurons has been extensively described[17,18]; however, the action of the potentially co-released neuromodulator SOM peptide from the SOM$^+$ cells is still not fully understood[19]. SOM has been shown to decrease excitatory neurotransmission and seizures. Reduced SOM levels in cortical areas positively correlate with cognitive deficits[20]. The only source of SOM within the cortical and hippocampal regions is the SOM$^+$ interneurons[19]; however, direct evidence of the somatostatinergic action of the SOM$^+$ cell population itself is lacking. In our study, we employed in vitro and in vivo electrophysiology combined with optogenetics and morphological investigations to study both the classical GABAergic and the less-studied somatostatinergic actions of dendrite-targeting SOM$^+$ interneurons in several layers of the MEC. We compared PV$^+$ and SOM$^+$ GABAergic cell actions and found striking differences in their target selectivity and postsynaptic timescales. PV$^+$ interneurons had a rapid inhibitory effect on all layers, whereas SOM$^+$ interneurons preferentially modulated the principal cells of layer$_{III–V}$ using fast GABAergic and slow somatostatinergic actions. In contrast, layer$_{II}$ stellate and pyramidal cells, both of which have been described as grid cells, showed predominantly GABAergic postsynaptic inhibition, mostly by fast spiking PV$^+$ interneurons. Optogenetic modulation of SOM$^+$ cell activity in awake behaving animals selectively disturbed specific short-term memory formation without affecting the ability of spatial memory retrieval.

## Results

**In the MEC, SOM$^+$ interneurons show strong layer-specific innervation, meanwhile PV$^+$ interneurons innervate all principal cells equally.** We induced ChR2 expression either in SOM$^+$ or in PV$^+$ interneurons in the MEC (Fig. 1a, b). First, we tested the light-evoked electrophysiological responses[21] in patch-clamp recordings from labeled GABAergic cells in both SOM-Cre-ChR2 and PV-Cre-ChR2 mice. In agreement with other reports[22,23], both SOM$^+$ ($n = 6$, $N = 5$) and PV$^+$ ($n = 9$, $N = 7$) interneurons showed narrow action potentials (APs; AP half-width: PV: $0.33 \pm 0.03$ ms; SOM: $0.39 \pm 0.05$ ms, $p = 0.39$) and fast, nonaccommodating firing in response to depolarizing currents (inter-spike interval: PV: $6.14 \pm 0.18$ ms, SOM: $8.24 \pm 1.54$ ms, accommodation index: PV: $1.58 \pm 0.13$, SOM: $1.55 \pm 0.11$). SOM-ChR2$^+$ and PV-ChR2$^+$ neurons showed similar spiking thresholds (SOM: $-46.4 \pm 2.1$ mV; PV: $-48.1 \pm 1.5$ mV). Both PV-ChR2$^+$ and SOM-ChR2$^+$ neurons elicited 1–3 APs as a response to brief (3 ms) light pulses (Fig. 1c, d). Therefore, we concluded that short light pulses generate comparable excitation/APs in both PV$^+$ and SOM$^+$ interneurons.

Next, we investigated the effect of SOM$^+$ interneuron innervation on multiple layers of the MEC. For this, we first checked the postsynaptic targets of mCherry-tagged SOM-ChR2$^+$ synaptic boutons in all investigated layers using electron microscopy. The boutons targeted spines or thin dendrites (layer$_I$: spines 25, dendrites: 15; layer$_{II}$: spines 28, dendrites: 17; layer$_{III–V}$: spines: 50, dendrites: 43; $N = 5$, Supplementary Fig. 1), but no somata were innervated by the SOM$^+$ interneurons.

We also recorded principal cells from different layers of the MEC in the SOM-ChR2 mice, in order to examine their light-evoked postsynaptic responses. A dense SOM immunoreactive axonal cloud and SOM-Cre-ChR2 axons (but no PV$^+$ boutons) can be found in layer$_I$ (Supplementary Fig. 1). However, SOM$^+$ interneurons elicited only moderate postsynaptic responses in layer$_{II}$ principal (stellate and pyramidal) cells located closely near layer$_I$. Both reelin$^+$ stellate ($n = 33$, $N = 18$) and WFS1/calbindin$^+$ pyramidal cells ($n = 9$, $N = 7$) responded only moderately (L$_{II}$ stellate: $2.2 \pm 0.38$ mV, L$_{II}$ pyramidal: $0.66 \pm 0.16$ mV) to light-evoked SOM$^+$ interneuron activation (Fig. 1e–g). The recorded cells showed dense dendritic arborizations surrounded by robust ChR2-mCherry expression from SOM$^+$ axons; thus the moderate postsynaptic effect was not due to truncated dendritic trees (Fig. 1e, f) or the low driving force of Cl$^-$ (CsCl intracell solution, Cl$^-$ equilibrium potential $\approx -27$ mV, see "Methods").

In contrast to layer$_{II}$, the pyramidal cells of layer$_{III}$ ($n = 15$, $N = 12$) and layer$_V$ ($n = 8$, $N = 6$) responded with a magnitude higher amplitude to the whole-field illumination of the same duration and intensity (L$_{III}$: $13.58 \pm 2.02$ mV, L$_V$: $16.6 \pm 2.09$ mV, Fig. 1f, g). Comparing the amplitudes of the light-induced postsynaptic potentials (PSPs) in the stellate cells with the PSPs in pyramidal cells of layer$_{II}$, as well as comparing the PSPs in layer$_{III}$ vs layer$_V$ pyramidal cells showed no differences; however, every other comparison (layer$_{II}$ cells vs layer$_{III}$ or layer$_V$ cells) showed notable differences ($p < 0.0001$, variance analysis with Tukey's multiple comparison test). Therefore, we concluded that SOM$^+$ preferentially innervates deep layers (layer$_{III–V}$) instead of superficial layer$_{II}$ principal (pyramidal and stellate) cells (layer$_{IV}$ in the MEC, also known as lamina dissecans, is considered a pyramidal cell-free thin layer between layer$_{II}$ and layer$_V$. The apical dendrites of pyramidal cells of layer$_{VI}$ do not enter layer$_{I–V}$[24] where the strong, local PV-ChR2 or SOM-ChR2 expression has been detected; therefore, we did not record the pyramidal cells of layer$_{VI}$).

We sought to map the cell type and layer specificity of PV$^+$ GABAergic inhibitory motifs in the MEC. In line with the previous reports[25], PV$^+$ boutons mainly targeted layer$_{II–V}$ cells

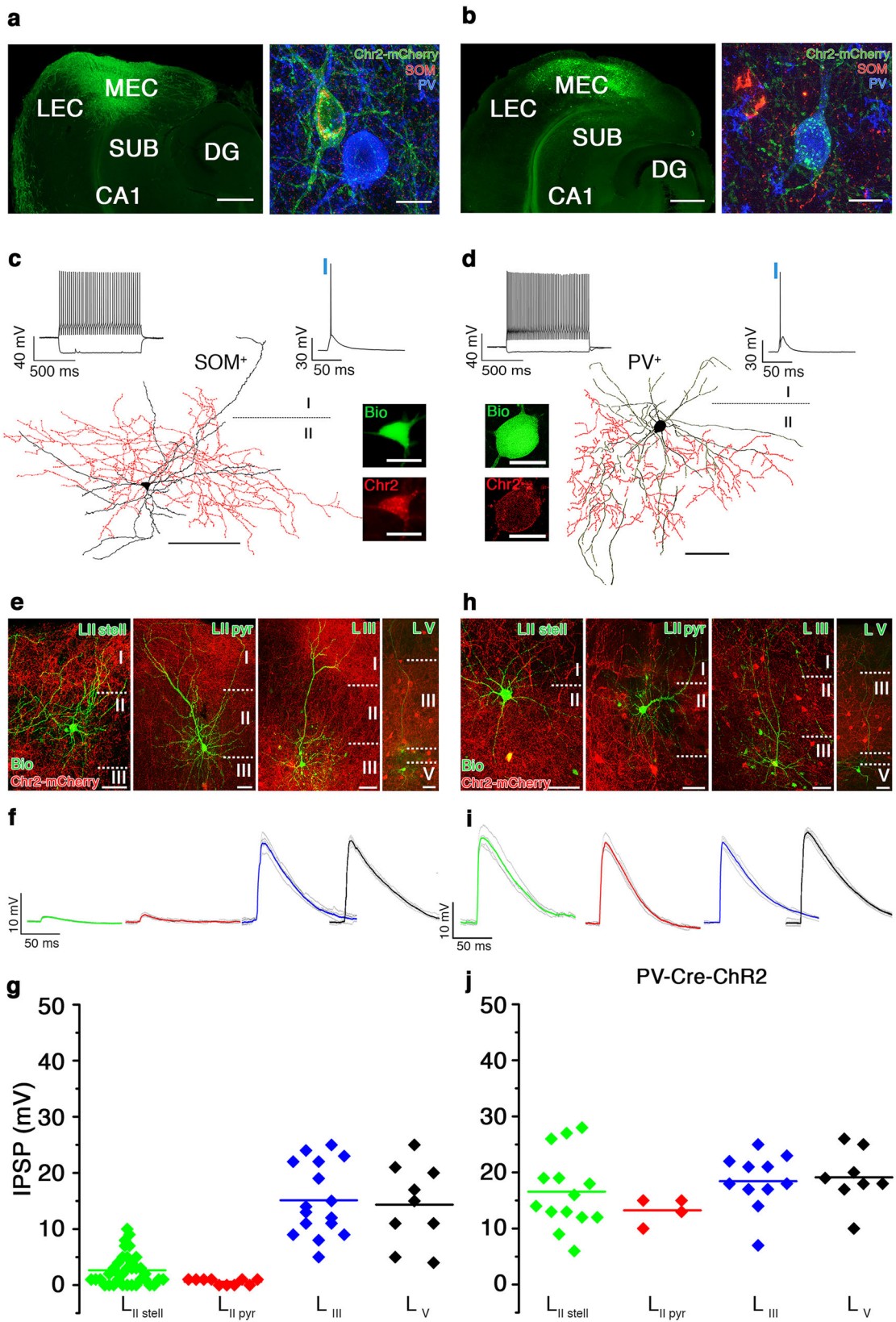

perisomatically (soma: 31, thick dendrite: 10, spine: 2, N = 2) and they did not target layer$_I$ (Supplementary Fig. 1). We applied short (3 ms), whole-field light illuminations[21] on the PV-ChR2-expressing slices and recorded the postsynaptic effects in principal (stellate and pyramidal, Fig. 1h) cells in different layers. Stellate (n = 14, N = 5 reelin$^+$ cells with prominent sag during

hyperpolarization[10,23,26]) and pyramidal cells (n = 4, N = 3 WFS1$^+$ or calbindin$^+$ [10,27]) responded with a fast, monosynaptic (2.17 ± 0.56 and 1.9 ± 0.6 ms delay time, respectively, p = 0.58) voltage change. These data are in agreement with previous reports investigating PV cell innervation onto layer$_{II}$ perisomatic-targeting neurons[11,23]. However, we found equally

**Fig. 1 SOM$^+$ local interneurons are strongly biased to innervate layer$_{III-V}$ pyramidal cells in the MEC. a, b** Left: low-magnification image of the horizontally sectioned temporal cortex of a SOM-Cre (**a**) and a PV-Cre (**b**) mouse showing Cre-dependent local ChR2 expression in the MEC. LEC lateral entorhinal cortex, SUB subiculum, DG dentate gyrus. Right: high-magnification image of ChR2-mCherry (green), SOM (red), and PV (blue) immunoreactive cells in the two Cre animals (scale bars: 250 and 10 μm). **c, d** Reconstructions of ChR2-expressing interneurons in the MEC of SOM-Cre (**c**) and PV-Cre (**d**) mice and the responses of the recorded cells to 1 s current injection (−200 and +200 pA) and to 3 ms (blue bar) photo-stimulation. Insets: confocal images of the biocytin-(Bio, green) filled interneurons showing the expression of Chr2-mCherry (red) (scale bars: 100 and 10 μm). **e, h** Confocal images of biocytin-filled layer$_{II}$ stellate (LIIstell), layer$_{II}$ pyramidal (LIIpyr), layer$_{III}$ (LIII), and layer$_{V}$ (LV) cells and the surrounding ChR2-mCherry-positive axons (red) in the MEC of SOM-Cre (**e**) and PV-Cre (**h**) mice (scale bars: 50 μm). **f, i** Whole-cell postsynaptic voltage response of the layer$_{II}$ stellate (green), layer$_{II}$ pyramidal (red), layer$_{III}$ (blue), and layer$_{V}$ (black) pyramidal cells, shown in **e** and **h**, to photo-stimulation of ChR2$^+$ interneurons (five superimposed consecutive traces in gray, averages in color) in SOM-Cre (**f**) and PV-Cre (**i**) mice, respectively. **g, j** Plots of the recorded events (IPSP, mV) in layer$_{II}$ stellate, layer$_{II}$ pyramidal, layer$_{III}$, and layer$_{V}$ pyramidal cells in SOM-Cre (**g**) and in PV-Cre (**j**) animals.

large postsynaptic voltage changes in layer$_{III}$ ($n = 11$, $N = 8$) and also in layer$_{V}$ ($n = 8$, $N = 3$) pyramidal cells as well. These inhibitory events were comparable with the effect on layer$_{II}$ principal cells (L$_{II}$ pyramidal: $13 \pm 1.08$ mV, L$_{II}$ stellate: $16.5 \pm 1.79$ mV and layer$_{III}$: $18.45 \pm 1.49$ mV, layer$_{V}$: $19.25 \pm 1.75$ Fig. 1i, j, $p = 0.23$, one-way analysis of variance (ANOVA) followed by Tukey's multiple comparisons test). Therefore, we concluded that, in contrast to previous immunohistochemical predictions[28], deep-layer (layer$_{III-V}$) pyramidal cells receive strong PV$^+$ innervation; thus PV$^+$ GABAergic cells have an overall strong GABAergic inhibition on all principal (pyramidal and stellate) cell types in layer$_{II-V}$ of the MEC.

The differences in SOM$^+$ interneuron innervation on principal cells of the deep vs superficial layers in the MEC may reflect a more general cortical/hippocampal microcircuit organization. In the somatosensory cortex, the layer specificity of distinct types of SOM$^+$ interneurons has been revealed[29], but the relative strengths of these inhibitory subtypes on pyramidal cells in supragranular, infragranular, and granular layers have not been compared. In CA1 of the hippocampus, superficial and deep pyramidal cells receive different level of PV$^+$ inhibition[9], but the relative strength of SOM$^+$ innervation has not been investigated. To examine this, we investigated the monosynaptic inhibitory effects of all types of SOM$^+$ interneurons in the somatosensory cortex and in the CA1 region (Fig. 2). We induced ChR2 expression in SOM$^+$ cells in the somatosensory cortex and in the dorsal hippocampus CA1 within the same SOM-Cre mouse line (SOM-Cre) and with the same AAV-ChR2 injection (see "Methods"). The postsynaptic effects of short, whole-field light illuminations on the ChR2-expressing slices were recorded in pyramidal cells in different layers of the somatosensory cortex (Fig. 2a, b). Layer$_{II-III}$ ($n = 7$, $N = 2$), layer$_{IV}$ ($n = 4$, $N = 3$), and layer$_{V-VI}$ ($n = 8$, $N = 3$) pyramidal cells showed similar latency (layer$_{II-III}$: $1.87 \pm 0.12$ ms, layer$_{IV}$: $2.21 \pm 0.39$ ms and layer$_{V-VI}$: $2.37 \pm 0.32$ ms, $p = 0.529$, Kruskal–Wallis test) and similar amplitude postsynaptic effects (layer$_{II-III}$: $3.43 \pm 0.72$ mV, layer$_{IV}$: $4.5 \pm 1.8$ mV, layer$_{V-VI}$ $3.4 \pm 0.65$, $p = 0.94$, Kruskal–Wallis test, Fig. 2c, d). In the dorsal hippocampus CA1, both superficial and deep pyramidal cells (Fig. 2e, f) received strong monosynaptic inhibition from SOM$^+$ interneurons (latencies: superficial: $2.5 \pm 0.22$ ms, deep: $2.41 \pm 0.26$; amplitudes: superficial: $3.04 \pm 0.21$ mV ($n = 9$), deep: $3.46 \pm 0.45$ mV ($n = 10$, $N = 9$), Fig. 2g, h, $p = 0.39$). Therefore, we concluded that the overall SOM$^+$ innervation in the somatosensory cortex and in the hippocampal CA1 region does not differentiate between superficial or deeper located pyramidal cells. Thus the preferential innervation of the deep-layer pyramidal cells in the MEC by SOM$^+$ GABAergic cells may suggest that they have specific roles in this brain area.

**SOM$^+$ interneurons inhibit pyramidal cells longer than PV$^+$ interneurons.** Subsequently, we tested whether the PV$^+$ and SOM$^+$ local inhibitory interneurons have different effects on the

firing activity of pyramidal cells in the MEC. First, we recorded the multi-unit activity of the MEC in head-fixed awake transgenic (PV-Cre-ChR2 and SOM-Cre-ChR2) mice. When 10 ms light pulses were applied to the MEC expressing ChR2 Cre-dependently, putative PV$^+$ or SOM$^+$ interneurons increased their firing rate (Fig. 3a, PV-ChR2: from $8.2 \pm 1$ to $50 \pm 6$ Hz, $n = 122$, $N = 6$ SOM-ChR2: from $3.5 \pm 1$ to $27.3 \pm 7$ Hz, $n = 36$, $N = 4$), and putative layer$_{III-V}$ pyramidal cells were inhibited (Fig. 3b). The duration of inhibition in PV-ChR2 animals was shorter than in SOM-ChR2 animals (50% recovery from inhibition: PV-Cre-ChR2: $37 \pm 1$ ms $N = 6$, $n = 136$, SOM-Cre-ChR2: $89 \pm 7$ ms $N = 4$, $n = 51$, $p < 0.0001$, Wilcoxon rank-sum test, Fig. 3b). The duration of inhibition in PV-Cre-ChR2 animals was comparable with what was reported in previous in vivo studies on the MEC[14]. However, the prolonged inhibition in SOM-Cre-ChR2 animals suggested an atypical underlying inhibitory mechanism.

**The neuromodulator SOM influences the firing probability of layer$_{III-V}$ pyramidal cells.** SOM as a neuromodulator has been shown to evoke hyperpolarization in CA1 pyramidal cells[19] and has been hypothesized to be released from synaptic boutons of SOM-expressing interneurons[30]. We have tested the subcellular localization of SOM in SOM$^+$ interneuronal cell bodies and their axon terminals in the MEC. Stimulated emission depletion (STED) microscopy revealed that SOM immunoreactivity is localized mostly within the nuclei-surrounding structures, most likely endoplasmic reticulum (Fig. 4a), and no overall cytoplasmic localization can be detected. Strong SOM$^+$ immunoreactivity, however, can be detected in bouton-like formations, which have been found to contain vesicular glutamic acid transporter (Fig. 4b) granular structures (synaptic vesicles). Importantly, the SOM immunoreactivity in mCherry/ChR2-expressing boutons of SOM$^+$ interneurons is associated with granular/vesicular structures on STED images (Fig. 4c). Moreover, electron microscopic investigation of mCherry/ChR2$^+$-labeled boutons in the MEC of SOM-Cre animals revealed that large-sized synaptic vesicles (signs of neuropeptide content[31]) can be detected in the symmetrical, spine- (Fig. 4d) or thin-dendrite-innervating boutons (not shown). Taken together, we hypothesize that MEC SOM$^+$ interneurons contain releasable SOM in their synaptic terminals.

Elongated inhibitory action of SOM$^+$ cells compared to PV$^+$ cells has been reported, for example, in the prefrontal cortex[7], but the underlying mechanisms have not been revealed. To understand the mechanisms causing this unexpected difference between PV$^+$ and SOM$^+$ inhibitory effect on layer$_{III-V}$ pyramidal cells, we determined first whether the kinetics of the postsynaptic currents evoked by PV$^+$ and SOM$^+$ interneurons were similar. Dendritic filtering of PSPs can cause slower kinetics[32]. In our experiments, however, the rise times (SOM: $5.96 \pm 1.22$ ms ($n = 11$, $N = 2$), PV: $5.77 \pm 0.99$ ms ($n = 11$, $N = 3$), $p = 0.91$) and decay times (SOM: $179 \pm 34.5$ ms, PV: $199.8 \pm 21.6$ ms, $p = 0.17$, Mann–Whitney test) of the two groups did not differ (Fig. 5a). This finding is

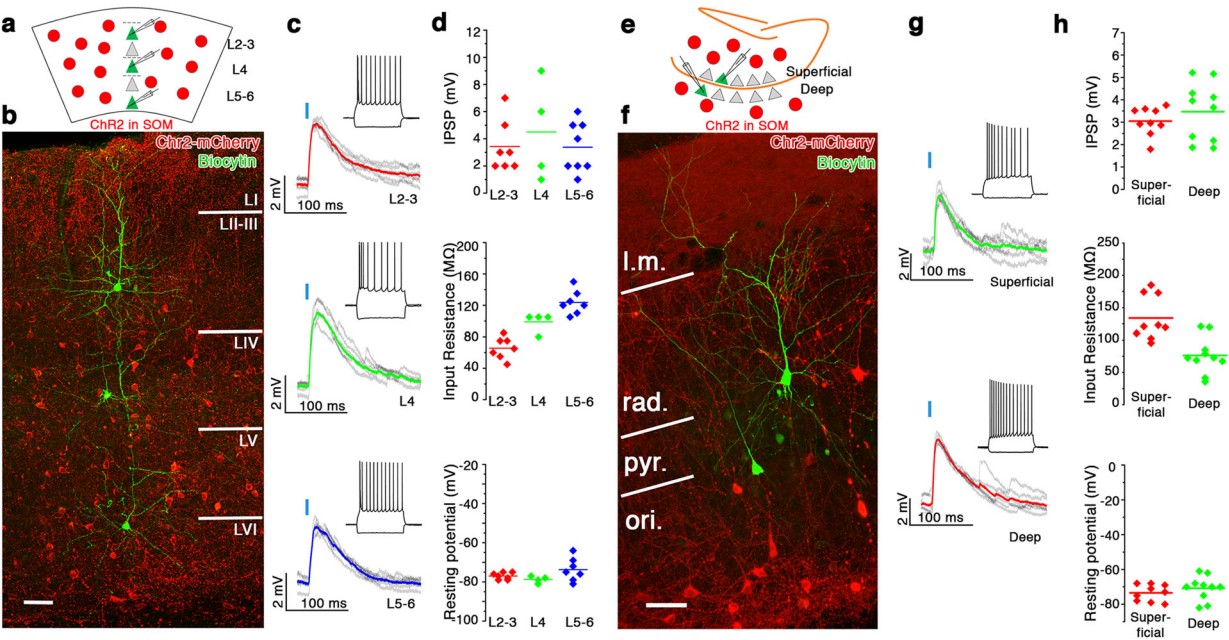

**Fig. 2 Pyramidal cells in different layers are equally innervated by SOM$^+$ interneurons in the somatosensory cortex and in the hippocampal CA1 region. a** Schematic of the experimental configuration in the somatosensory cortex: brief (3 ms) whole-field photo-stimulation of ChR2-expressing SOM$^+$ cells (red circles) and simultaneous recordings from neighboring pyramidal cells from several layers. **b** Overview image of three recorded and biocytin-filled pyramidal cells in layer$_{III}$, layer$_{IV}$, and layer$_{VI}$ (green) in SOM-ChR2 (red) expressing somatosensory cortex (coronal section). **c** Light-evoked (blue lines represent time of illumination) postsynaptic potential changes in the three recorded cells. Colored lines are the averages of individual (gray) events. Insets: responses to hyperpolarizing and depolarizing current steps of the recorded cells. **d** Postsynaptic potential (IPSP, top), input resistance (middle), and resting membrane potentials (bottom) of the layer$_{II-III}$ (red), layer$_{IV}$ (green), and layer$_{V-VI}$ (blue) cells. **e** Schematic figure of the experimental configuration in the hippocampal CA1 region. **f** One superficial (top right) and one deep pyramidal cell (bottom left, green) surrounded by ChR2-mCherry-expressing cells in SOM-ChR2 animal (coronal section). l.m. lacunosum moleculare, rad. stratum radiatum, pyr. stratum pyramidale, ori. stratum oriens. **g** Postsynaptic responses of the two recorded cells to 3 ms photo-stimulations. Colored lines are the averages; gray lines are the individual events. Insets: responses to hyperpolarizing and depolarizing current steps of the recorded cells. **h** Postsynaptic potential (top), input resistance (middle), and resting membrane potentials (bottom) of the superficial (red) and deep (green) pyramidal cells. Note that the intracellular solution contained 40 mM CsCl solution, producing depolarizing effect of GABAA receptor opening. L.m. stratum lacunosum moleculare, rad stratum radiatum, pyr stratum pyramidale, ori stratum oriens. Scale bars: 50 μm.

in agreement with that of the comparison of the basket cell and Martinotti cell inhibitory PSP kinetics[33]. Moreover, applying the GABAA receptor blocker gabazine completely eliminated the postsynaptic effect of both PV$^+$ and SOM$^+$ interneurons (Fig. 5b and Supplementary Fig. 1e); thus no GABAB receptor activation was involved in the prolonged postsynaptic action of SOM$^+$ interneurons.

What is causing this prolonged inhibitory action of SOM$^+$ interneurons, if neither dendritic filtering nor the GABAB receptor is involved in the mechanism? SOM has a potential neuromodulator effect, and it has been hypothesized that it is released from SOM$^+$ axon terminals[19,30] and is located in GABAergic axon terminals in the MEC (see above). SOM has been reported to have a potential regulatory effect on cortical excitability[34]. Therefore, we investigated whether the putative effect of SOM can be eliminated or mimicked experimentally. Unfortunately, there is no specific SOM antagonist; therefore, we crossed SOM-Cre animals with STT4KO animals (SST4 is strongly expressed in the hippocampus[35]) and induced ChR2 expression in the MEC SOM$^+$ interneurons. First, we compared the durations of the light-induced inhibition in SOM-ChR2, SST4 KO, and PV-ChR2 MEC layer$_{III-V}$ pyramidal cells in vitro. We held the recorded cell's membrane potential at the level where low-frequency firing (inter-spike interval: SOM $140 \pm 16$ ms, PV $127 \pm 7$ ms, SST4 KO $126 \pm 6$ $p = 0.7938$, Kruskal–Wallis test) occurred and excited the SOM-ChR2$^+$ or PV-ChR2$^+$ cells/axons with light pulses (Fig. 5c). The firing of principal cells recovered

much earlier after the excitation of PV$^+$ interneurons and in the SST4 KO animals than in the SOM-ChR2 animals (PV $220 \pm 17$ ms, $n = 16$, $N = 2$; SST4 KO $240 \pm 21$ ms, $n = 15$, $N = 2$; SOM $340 \pm 32$ ms, $n = 10$, $N = 3$; $p = 0.0023$, SST4 KO vs PV is not significant (alpha $= 0.05$), one-way ANOVA followed by Tukey's multiple comparisons test, Fig. 5d).

We found that the inhibitory effect of SOM$^+$ cells in SST4 KO animals was shorter than in wild-type animals in vivo as well (Fig. 5e). Head-restrained awake SST4 KO animals expressing ChR2 in MEC SOM$^+$ cells were excited by 10 ms light pulses, similar to the SOM-ChR2 and PV-ChR2 mice experiments; the 50% return time of the firing probability was shorter than in SOM-ChR2 animals (SST4 KO: $53 \pm 3$ ms $N = 3$, $n = 64$; SOM-Cre-ChR2: $N = 4$, $89 \pm 7$ ms $n = 51$, mean rank difference: 43.41, $p < 0.01$, Fig. 5f) but longer than in PV-Cre-ChR2 animals ($37 \pm 1$ ms $N = 6$, $n = 136$; mean rank difference: 49.18, $p < 0.01$; and mean rank difference for SOM-Cre-ChR2 vs PV-Cre-ChR2: 92.59, Kruskal–Wallis test followed by post hoc Dunn's pairwise comparison tests), indicating the involvement of SST4 receptors in the prolongation of inhibitory action of SOM$^+$ interneurons on layer$_{III-V}$ pyramidal cells. However, other factors (e.g., other SST receptors) may also contribute, since the duration of inhibition in SST4 KO animals was longer than in PV-ChR2 animals. Subsequently, we tested the effect of the SOM receptor agonist J-2156[34] on the firing properties of layer$_{II-V}$ principal cells. This agonist only had a small effect on the firing frequency of layer$_{II}$ principal cells ($19.8 \pm 1.9$ Hz control vs $18.4 \pm 2.4$ Hz J-

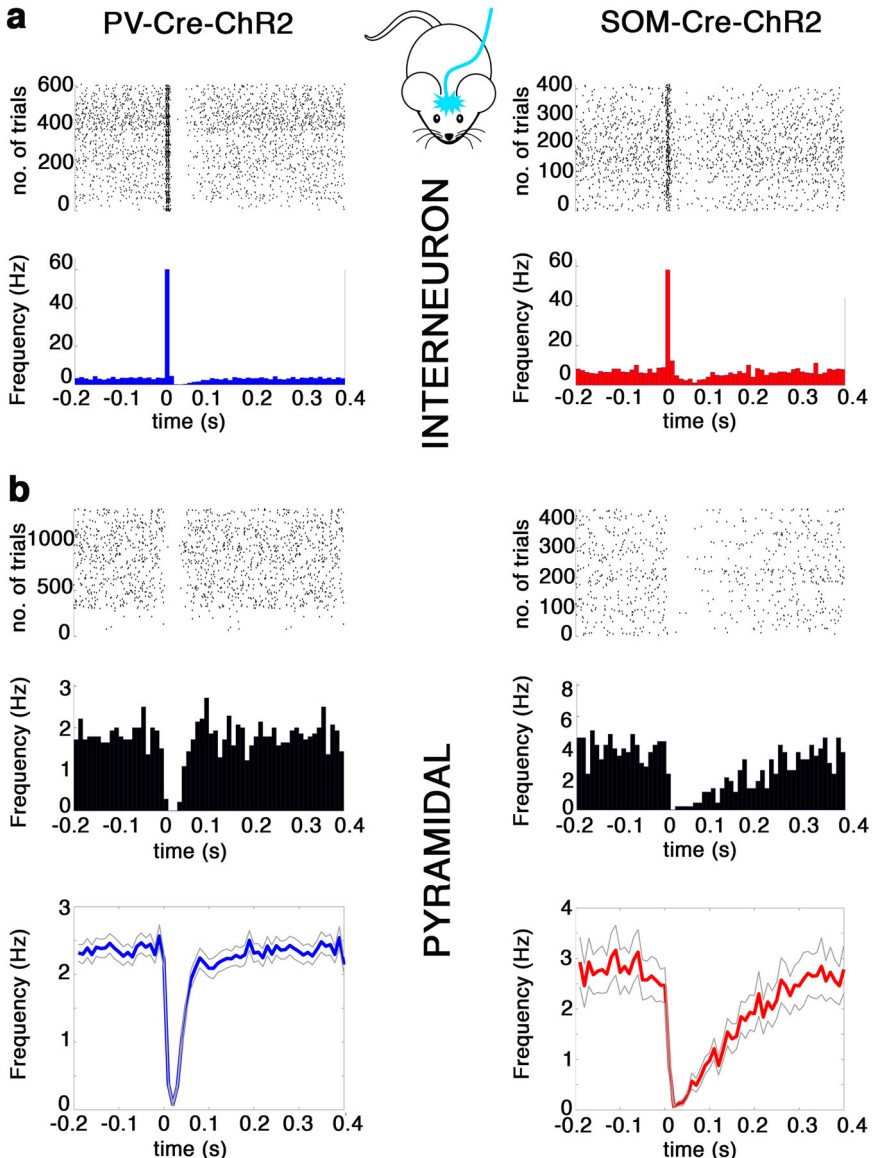

**Fig. 3 Prolonged SOM+ inhibition in the MEC in awake behaving mice.** Optogenetic modulation of MEC networks in vivo in PV-Cre-ChR2 (left column) and SOM-Cre-ChR2 (right column) animals. **a** Representative spike raster (top) and peri-stimulus histogram (PSTH, bottom) of light-responsive putative PV+ (left) and a SOM+ (right) interneuron aligned to the 10 ms light onset (0 s). Middle: schematic representation of experimental set-up. **b** Raster plot (top) and PSTH (middle) of representative light-inhibited putative pyramidal cells in PV-Cre (right) and SOM-Cre (left) animals. Bottom: average (solid line) and standard error of the mean (gray) of all inhibited putative pyramidal cells in PV-Cre (left) and SOM-Cre (right) animals. Note the quick return of firing in PV-Cre-ChR2 animals and the elongated inhibition in SOM-Cre-ChR2 animals. All traces aligned to light onset (0 s).

2156 @300 pA injected current, $n = 14$, $p = 0.0121$, paired $T$ test, Fig. 5g, h) but substantially decreased the firing frequency in layer$_{III-V}$ pyramidal cells ($12.9 \pm 0.9$ Hz control vs $9.9 \pm 0.9$ Hz J-2156 @150 pA, $n = 23$, $p < 0.0001$, paired $T$ test, Fig 5g, h). Notably, the layer specificity of SOM action follows the specificity of monosynaptic targets of SOM+ interneurons described above. Due to the lack of specific SOM receptor blockers[35], we could not directly ascertain whether the prolonged inhibitory effect of SOM+ cells can be shortened by SOM receptor blockers.

**SOM+ interneurons regulate short-term memory formation.** SOM-expressing interneurons have been reported to not effect grid-cell activity; instead, temporary suppression of this cell population increased the firing of cells with non-spatial or non-periodic spatial selectivity[8]. Layer$_{III-V}$ pyramidal cells have been suggested to play a major role in short-term memory

formation[15,16]; therefore we tested whether the activation of SOM+ cells during the exploratory behavior of mice can disturb working memory formation via the revealed strong specific inhibition of the layer$_{III-V}$ pyramidal cells. For this, we allowed SOM-Cre-ChR2 ($N = 9$), PV-Cre-ChR2 ($N = 5$), and SOM-Cre-EGFP ($N = 4$) with bilateral MEC ChR2 expression to explore the Y-maze. In this test, the naturally occurring behavior, which requires short-term memory formation, is the alternation of arms where the animal enters. We compared the running sessions with and without bilateral light excitation of the MEC (Fig. 6a). The spontaneous alternation score (spontaneous alternation performance (SAP)) was reduced in conditions when SOM+ cells were excited ($61.66 \pm 17.98$ vs $44.73 \pm 3.67\%$, $p = 0.0199$, paired $T$ test, Fig. 6b and Supplementary Video). SOM-EGFP ($46.71 \pm 22.34$ vs $58.34 \pm 8.39\%$, $p = 0.625$ Wilcoxon matched-pairs signed rank test) and PV-ChR2 ($56.95 \pm 9.26$ vs $56.05 \pm 12.55\%$, $p > 0.999$,

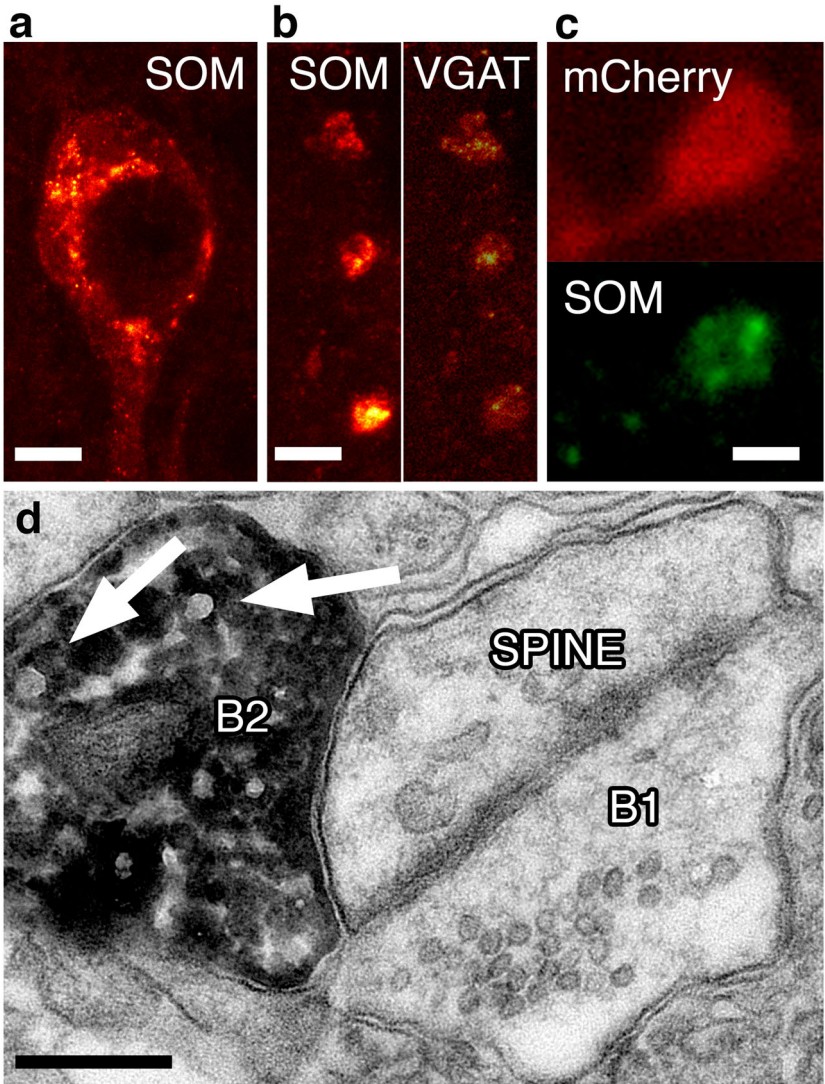

**Fig. 4 The SOM neuromodulator peptide is synthetized in the somatic region and packed into synaptic vesicles. a** STED image of SOM immunoreactivity in a soma in the MEC. Note the granular labeling of putative endoplasmatic reticulum surrounding the empty nucleus. Scale: 2 µm. **b** SOM (left) and VGAT immunoreactivity detected with STED microscopy within the same synaptic boutons in MEC. Scale: 1 µm. **c** STED images of a Cre-dependent mCherry/ChR2-expressing bouton in the MEC of a SOM-Cre animal (top, red). SOM immunoreactivity in the same bouton shows granular, putative synaptic vesicle localization (bottom, green). Scale: 0.5 µm. **d** Electron microscopic image of a spine in layer$_{II}$ of the MEC innervated by two different type of boutons. B1 is a putative excitatory bouton with similar-sized synaptic vesicles. B2 is a bouton of mCherry/ChR2 (DAB precipitate) expressing SOM$^+$ interneuron. Note the larger-sized synaptic vesicles (arrows) occurring among the normal-sized vesicles. Scale: 200 nm.

Wilcoxon matched-pairs signed rank test) animals did not show reduced spontaneous alternation scores during the light activation sessions.

To determine whether the decreased spontaneous alternation in SOM-ChR2 animals was due to the inability to retrieve memory traces, we have trained the animals to find the hidden platform in the Morris water maze. After 5 days of training, on the probe trial day the escape latency was measured in the SOM-ChR2, PV-ChR2, and SOM-EGFP animals (Fig. 6c, d). Short light pulses were applied to the MEC (while the animals were performing the test, see "Methods"). No decrease in escape latency was detected in any of the investigated groups (SOM-ChR2, $N = 8$, $49.25 \pm 18.46$ s vs $37.79 \pm 21.17$ s, $p = 0.375$, Wilcoxon matched-pairs signed rank test; SOM-EGFP $N = 4$, $26.9 \pm 24.06$ s vs $16.25 \pm 18.15$ s, $p = 0.875$, Wilcoxon matched-pairs signed rank test, PV-ChR2 $N = 5$, $36 \pm 22.59$ s vs $26.14 \pm 10.12$ s, $p = 0.625$, Wilcoxon matched-pairs signed rank test, Fig. 6d). These data indicate that dendritic inhibition of layer$_{III-V}$

pyramidal cells conveyed by SOM$^+$ local interneurons influences short-term memory formation but does not block the retrieval of already learned locations.

## Discussion

In the present study, we have characterized the layer-selective GABAergic and somatostatinergic actions of SOM-expressing interneurons and compared their actions on local principal cells in the MEC with PV-expressing local interneurons. The deep layer (layer$_{III}$ and layer$_V$) pyramidal cells received strong monosynaptic inhibition on their dendrites from SOM$^+$ interneurons and ceased their firing longer than after PV$^+$ inhibition. We have shown that the PV$^+$ interneurons had strong effects on layer$_{II}$ stellate and pyramidal cells, as well as layer$_{III}$ and layer$_V$ pyramidal cells, whereas SOM$^+$ interneurons were highly biased to innervate layer$_{III}$ and layer$_V$ pyramidal cells and innervate layer$_{II}$ principal cells only moderately. This layer specificity of SOM$^+$

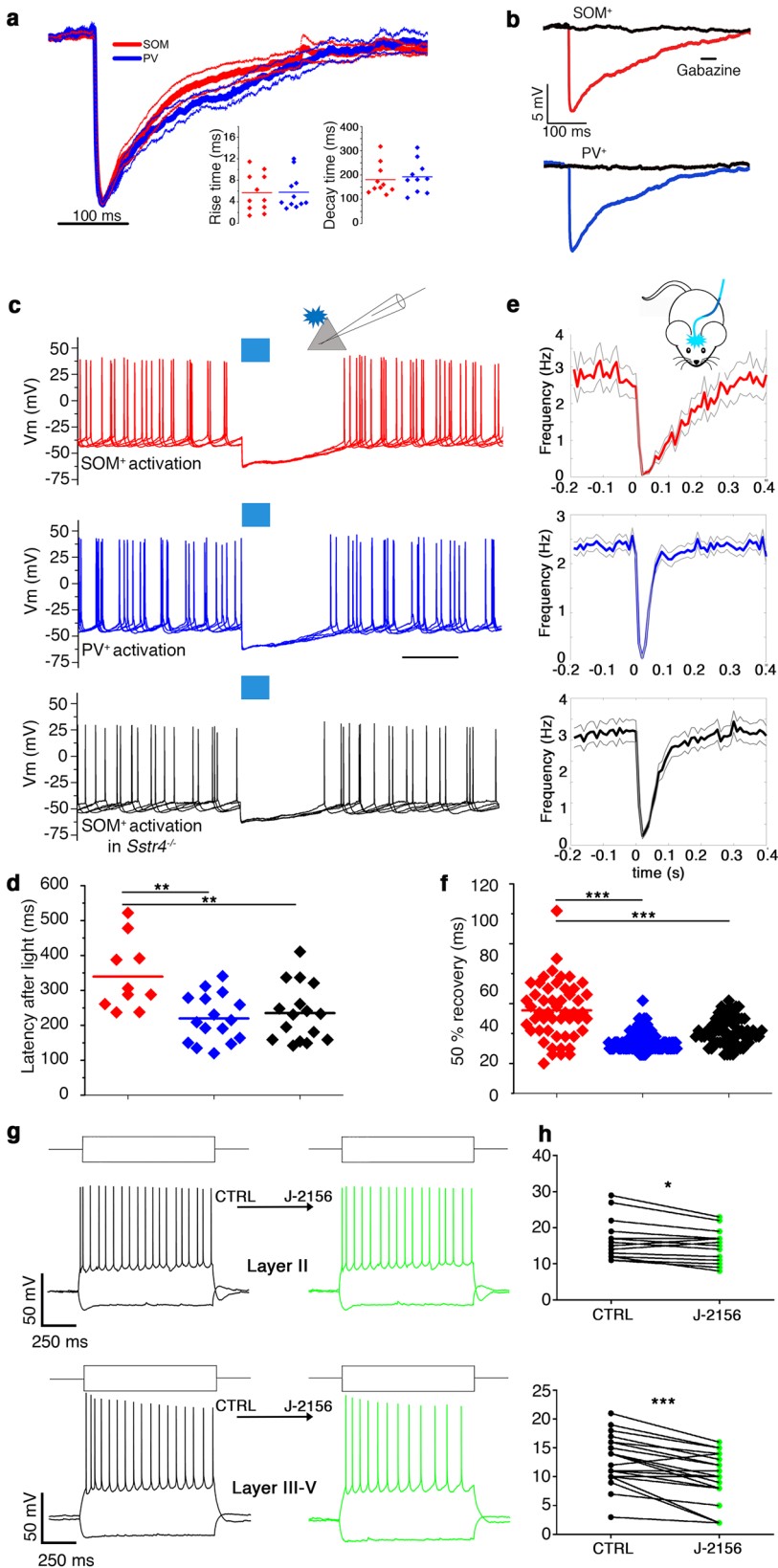

interneurons seems to be MEC specific, since neither CA1 in the hippocampus nor the somatosensory cortex showed inter-layer differences. However, the same layer specificity may be present in the lateral entorhinal cortex, which resembles the cytoarchi-tecture[36]. We have also revealed that the action of SOM as a neuromodulator seems to demonstrate the target specificity of

SOM⁺ interneurons; the layer$_{III}$–layer$_V$ pyramidal neurons showed a prominent decrease in firing frequency when the SST4 receptor agonist J-2156 was applied. Considering that the SST4 receptor mainly acts positively on M-currents[35], our results are in agreement with the proposal that, under the M-current block, the pyramidal cells excitability in MEC layer$_{III–V}$ increases[37]. A

**Fig. 5 Prolonged inhibition in layer$_{III-V}$ pyramidal cells is mediated by the neuromodulator peptide SOM. a** Decay and rise times similarities of PSPs elicited by PV$^+$ and SOM$^+$ interneurons indicate classical fast GABAA receptor-mediated inhibition. The averages (thick lines) and S.E.M. (thin lines) of PSPs after short (3 ms) light pulses in PV-ChR2 (blue) and SOM-ChR2 (red) MEC. Layer$_{III-V}$ pyramidal cells held at resting membrane potentials. Inset: rise and decay times of individual pyramidal cells in PV-ChR2 (blue) and SOM-ChR2 (red) animals. **b** Both in SOM-ChR2 and PV-ChR2 animals, the PSPs (red, blue lines, respectively) can be completely eliminated by gabazine (black traces). **c** Action potentials (elicited by depolarization) in layer$_{III-V}$ pyramidal cells are stopped by 100 ms light pulse for different time in SOM-ChR2 (red), PV-ChR2 (blue), and SOM-ChR2+/SST4 KO (black) animals. **d** Statistic showing MEC deep principal cells firing latencies (as shown in **c**) after optogenetic stimulation of SOM-ChR2 (red), PV-ChR2 (blue), and SOM-ChR2+/SST4 KO (black) animals. **e** Population averages of the light-inhibited pyramidal cells in in vivo awake mice. Note that plots for SOM$^+$ (red) and PV$^+$ (blue) are same as in Fig. 3b for comparison with SST4 KO animals (black). **f** Statistics showing 50% recovery times of firing after light-induced inhibition in SOM$^+$ (red), PV$^+$ (blue), and SST4 KO (black) animals. **g** Representative voltage responses of MEC layer$_{II}$ (up) and deep layer (bottom) principal cell upon step current injections (150 and −200 pA) under control conditions (left) and after bath application of 1 μM J-2156 (right, green). **h** Firing frequencies of MEC layer$_{II}$ (up) and deep layer (bottom) principal cells upon current injections under control and J-2156-treated conditions. *$p < 0.05$, **$p < 0.01$, ***$p < 0.001$, paired Student's $T$ test for in vitro and Wilcoxon rank-sum test for in vivo experiments.

limitation of the present study is that the specific cell type and subcellular localization of SOM receptors are not known. SST4 is the most widely expressed SOM receptor in the brain[38]; therefore, we used SST4 KO animals in order to show the involvement of SOM as a neuromodulator for the prolonged inhibitory effect of SOM$^+$ GABAergic interneurons. However, we cannot rule out the involvement of other SOM receptors such as SST2, which also can be found in several brain areas[38]. The inhibitory effect of SOM$^+$ interneurons lasted shorter in the SST4 KO animals than in SST4 receptor-containing animals; however, it lasted longer than the inhibitory effect of perisomatically targeting PV$^+$ interneurons. This might indicate the involvement of other SOM receptors or other mechanisms that prolong the inhibition of the pyramidal cells. However, the GABAB receptor is not likely to contribute to this prolonged inhibition (GABAA receptor blocker fully eliminated the postsynaptic effects of light pulses in the principal cells of SOM-ChR2 MEC). Another limitation of the present study is the lack of direct behavioral experiments proving the contribution of SOM receptors to the modulation of short-term memory formation. We could not use SST4 KO animals in our behavioral experiments due to major anxiety and depression phenotype differences compared to wild-type animals. This may be caused by the high expression level of SST4 receptor in the amygdala[39].

The prolonged inhibitory effect of SOM$^+$ local GABAergic cells on pyramidal cells has been shown in the prefrontal cortex[7]; therefore, we assume that the elongated effect of SOM$^+$ interneurons may occur in several brain areas. The different actions of perisomatically targeting PV$^+$ and dendritically targeting SOM$^+$ cells in local networks have been revealed in several brain areas. Perisomatic inhibition is considered to entrain rather than inhibit principal cell activity. For example, during high-frequency ripple events in the hippocampus, both pyramidal cells and fast-spiking, perisomatic inhibitory PV$^+$ basket cells show elevated activity[1,40]. The suppression of dendritic inhibition, on the other hand, has been shown to increase the firing rate and bursting[2,4,41] of pyramidal cells.

Both PV$^+$ and SOM$^+$ interneurons have several subtypes. PV is expressed in basket and axo-axonic cells[1]. SOM$^+$ cells are also largely heterogeneous: Martinotti and non-Martinotti cells send axon arborisations to different layers in the neocortex[29]. The potential layer and cell-type specificity of PV$^+$ cells[9] may be masked in our experimental set-up. However, we demonstrated a difference in SOM$^+$ dendritic innervation of the principal cells of layer$_{II}$ and layer$_{III-V}$, which is specific to the MEC. The mechanism underlying the innervation of stellate cells in layer$_{II}$ and pyramidal cells in different layers by other dendrite-targeting cells, such as neurogliaform cells[42], has not been investigated in the current study.

The importance of temporal and spatial organization of GABAergic inhibitions converging on principal cells during behaviorally relevant brain oscillations, memory formation, and consolidation processes has been highlighted in several brain areas[43–45]. In the MEC, inhibitory inputs have been suggested to play a major role in synchronizing events[44,46] and in grid-cell formation[8,11,47]. Network models of grid-cell firing involve fast-spiking perisomatic-targeting GABAergic cells reciprocally connected with excitatory grid cells. These attractor dynamic models are based on the finding that principal cells in layer$_{II}$ communicate with each other mostly via PV$^+$ fast-spiking interneurons[11] (but see refs. [13,48]). Optogenetically tagged PV$^+$ interneurons in the MEC show moderate spatial selectivity[14]; moreover, some network models predict grid-cell and inverted grid-cell behavior of fast spiking interneurons[12]. This prompts the exploration of other network motifs, which may fuel more accurate modeling of grid-cell activity.

In addition to spatial navigation, MEC is also engaged in memory formation processes. The pyramidal cells in layer$_{III-V}$ are dedicated to holding temporary information during memory delays[15,16]. This memory process is enabled by the unique property of layer$_{III-V}$ pyramidal cells; they are capable of maintaining high-frequency firing even after their initial input has ceased[15]. Although this transient information storage can be rapidly depleted by massive hyperpolarization[15], a circuit-based inhibitory mechanism that can enable this phenomenon has not been revealed. Here we have shown that the stimulation of SOM$^+$ local GABAergic interneurons disturbed short-term memory-related behavior of the animals, whereas the stimulation of PV$^+$ interneurons did not. However, the effectivity of light stimulation of ChR2-expressing interneurons may differ between animals.

Our results reveal that the GABAergic and somatostatinergic inhibition elicited by SOM$^+$ local interneurons show strong layer preference in MEC. The activities of layer$_{III-V}$ pyramidal cells, which have important implications for handling memory, are heavily influenced by SOM$^+$ inhibition. Layer$_{II}$ principal cells, however, receive only moderate level of inhibition from local SOM$^+$ interneurons. This sparse inhibition may have a permissive role in the dendritic encoding of converging spatial information in principal cells in layer$_{II}$ of MEC. Future experiments addressing the morphological and molecular identity of interneurons and the effect of their manipulation is awaited in order to shed light of other MEC-specific microcircuit motifs.

## Methods

**Experimental animals and viral injection**. The experiments were approved by the Ethics Committee on Animal Research of Pecs, Hungary (license #: BA02/2000-1-2015). Male and female 4–6-week-old SOM-Cre (Sst-IRES-Cre, Stock No.: 013044, The Jacksons Laboratory), PV-Cre (B6;129P2-Pvalb$^{tm1(cre)Arbr}$/J, Stock No.: 017320 The Jacksons Laboratory), and SOM-Cre crossed with SST4 KO[49] mice were used for the experiments. The animals were housed in 12 h light/12 h dark

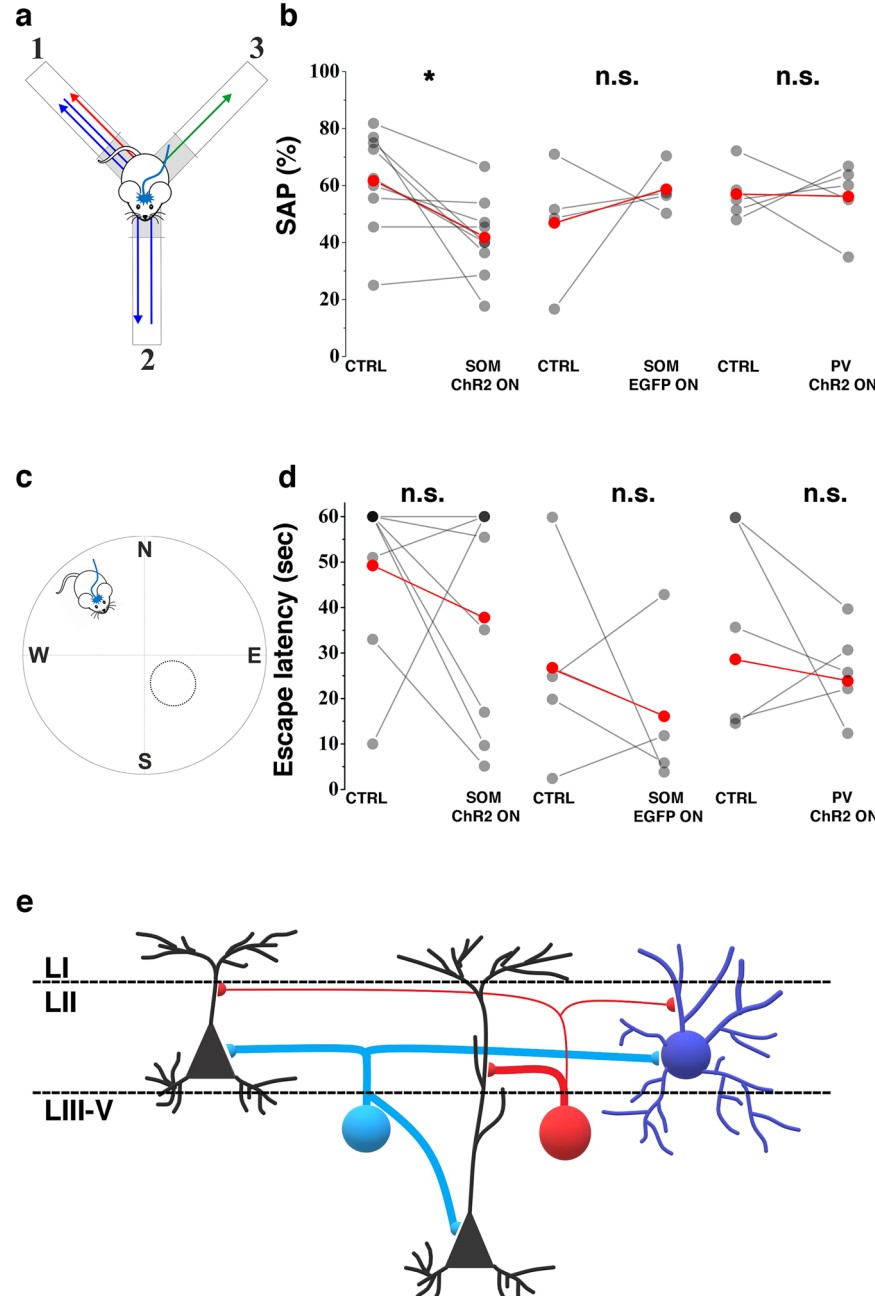

**Fig. 6 The inhibition of deep layer cells by SOM$^+$ GABAergic interneurons regulates short-term memory formation without influencing spatial navigation. a** Schematic indicating the design of the Y-maze experiment combined with optogenetic stimulation of SOM$^+$ cells in the MEC. Light was illuminated into the MEC during exploration, when animal was in the center area (gray). Alternation was considered correct, when after two entries (e.g., arms #1, #2) the animal entered the unvisited (#3) arm. **b** Correct spontaneous alternations of ChR2-expressing SOM-Cre (left), EGFP-expressing SOM-Cre (middle), and ChR2-expressing PV-Cre animals (right) during control (CTRL) and during ChR2 exciting light (ON) (*$p < 0.05$, paired Student's $T$ test). **c** Schematic representing the Morris water-maze experiments. Light pulses exciting ChR2$^+$ SOM cells were applied while the animal was finding the hidden platform on the last day (day 6) of the trainings. **d** Escape latencies of ChR2-expressing SOM-Cre (left), EGFP-expressing SOM-Cre (middle), and ChR2-expressing PV-Cre animals (right) during control (CTRL) and during light pulses (ON). **e** Schematic summarizing the network motif revealed in the present study. Layer$_{II}$ pyramidal (black, left), layer$_{II}$ stellate (black, right), layer$_{III-V}$ pyramidal (black, bottom), PV$^+$ interneuron (PV, blue), and SOM$^+$ interneuron (SOM, red). Thicker axons represent stronger inhibition on the targeted cells.

cycle. Animal handling was performed according to the regulations of the European Community Council Directive and approved by the local ethics committee. The animals were deeply anesthetised (isoflurane, 4% initial dose then 1% during the surgery). Small craniotomy was drilled in the skull above the MEC (4 mm lateral and 8 mm ventral from bregma). To selectively express ChR2 in SOM$^+$ or PV$^+$ interneurons, adeno-associated virus vector coding ChR2-mCherry fusion protein under the CBA promoter (AAV9.EF1.dflox.hChR2(H134R)-mCherry. WPRE.hGH (Addgene20297), Penn Vector Core, University of Pennsylvania,

USA) was injected 2.5–3.5 mm ventral from craniotomy (40–70 nl of undiluted, ~$10^{12}$ GC/ml) at postnatal day P25–30 into the MEC. Mice were sacrificed 2 weeks post-injection for slice preparation.

For behavioral experiments, bilateral MEC injection and, for head-fixed silicon probe recording experiments, unilateral MEC injection were performed. Optical fibers (200 µm, 0.39 NA, Thorlabs) were implanted into the MEC in animals performing behavioral experiments. The light was delivered through a light-weight optical fiber (FT200EMT, Thorlabs) from the 465 nm light-emitting diode.

**Slice preparation**. Optogenetic ChR2 experiments were performed in acute horizontal slices taken from SOM-Cre or PV-Cre mice at P40–45 that were previously injected intracranially with a recombinant adeno-associated viral construct (see above). Under deep isoflurane anesthesia, mice were decapitated and horizontal slices for MEC and coronal slices for hippocampus and somatosensory cortex recordings (300 μm thick) were cut in ice-cold external solution containing (in mM): 93 NMDG, 2.5 KCl, 25 Glucose, 20 Hepes, 1.2 NaH$_2$PO$_4$, 10 MgSO$_4$, 0.5 CaCl$_2$, 30 NaHCO$_3$, 5 L-ascorbate, 3 Na-Pyruvate, 2 thiourea bubbled with 95% O$_2$ and 5% CO$_2$. Slices were transferred to artificial cerebrospinal fluid (ACSF) containing (in mM) 2.5 KCl, 10 glucose, 126 NaCl, 1.25 NaH$_2$PO$_4$, 2 MgCl$_2$, 2 CaCl$_2$, 26 NaHCO$_3$ bubbled with 95% O$_2$ and 5% CO$_2$. After an incubation period of 10 min at 34 °C in the first solution, the slices were maintained at 20–22 °C in ACSF until use. After recordings, the sections were immersed into fixative (4% paraformaldehyde 0.1% picric acid in 0.1 M phosphate buffer (PB)) for overnight fixation.

**In vitro electrophysiological recordings**. Patch pipettes were pulled from borosilicate glass capillaries with filament (1.5 mm outer diameter and 1.1 mm inner diameter; Sutter Instruments) with a resistance of 2–3 MΩ. The pipette recording solution contained (in mM) "CsCl containing intracell": 3.5 KCl, 40 CsCl, 90 K-gluconate, 1.8 NaCl, 1.7 MgCl$_2$, 0.05 EGTA, 10 Hepes, 2 Mg-ATP, 0.4 Na$_2$-GTP, 0.2% Biocytin; "low chloride intracell": 5 KCl, 135 K-gluconate, 1.8 NaCl, 0.2 EGTA, 10 HEPES, 2 Na-ATP, 0.2% Biocytin, pH 7.3 adjusted with KOH; 290-300 mOsm. Whole-cell recordings were made with Axopatch 700B amplifier (with Clampex 10.7 and Axoclamp1.1, Molecular Devices) using an upright microscope (Nikon Eclipse FN1, with ×40, 0.8 NA water immersion objective lens) equipped with differential interference contrast (DIC) optics and fluorescence excitation source (CoolLED). DIC and fluorescence images were captured with an Andor Zyla 5.5 sCMOS camera. All recordings were performed at 32 °C in ACSF bubbled with 95% O$_2$ and 5% CO$_2$. In some experiments, 1 μM gabazine (Sigma-Aldrich) or 1 μM J-2156 (Tocris) were applied in the bath solution. The membrane potential was maintained at −70 mV in voltage clamp mode or adjusted close to that level with current injection (<100 pA) in current clamp mode. Cells with <20 MΩ access resistance (continuously monitored) were accepted for analysis. Signals were low-pass filtered at 5 kHz and digitized at 20 kHz (Digidata 1550B, Molecular Devices). The light (full field, 490 nm peak, 3 ms, CoolLed) was flashed on the slices through the immersion objective lens. In order to normalize the ChR2 expression level variability between animals, light power was manually adjusted until evoked saturated responses in the first recorded layer$_{III–V}$ pyramidal cell in the given animal. This initial light power was kept during the recording sessions on all the slices from the same animal. Accommodation index was calculated by dividing the average of the first two inter-spike intervals with the average of the last two inter-spike intervals during a 1-s-long depolarization step[50]. Firing frequency is calculated by counting all spikes during the 1-s-long depolarization square pulse. In vitro data analysis was performed with the help of Clampfit 10.7 (Molecular Devices) and Origin 8.6 (OriginLab Corporation).

**In vivo electrophysiological recordings**. All surgery was done under deep isoflurane anesthesia. Head bar implantation, acclimation, and craniotomy were done as described earlier[51]. Briefly, head bar was placed and attached with dental cement to the skull so that both dorsal hippocampal (2 mm lateral, 2 mm posterior from Bregma) and entorhinal cortex (4 mm lateral, 8 mm posterior from bregma) recording could be performed. Animals were trained to run or rest on an 8-inch spherical treadmill. On the day of the experiment, two 1 mm craniotomies were performed above the dorsal hippocampus and the entorhinal cortex on the same side. A linear silicon probe with 32 recording sides (A1x32-Poly2, Neuronexus) was placed into the CA1 region of dorsal hippocampus. The MEC was targeted with a 2 or 3 shank silicon probe. Each shank contained 8 recording sides with 50 μm spacing (Buzsaki32 design, Neuronexus). A 200-μm-thin optical fiber (200 μm, 0.39 NA) was attached to the probe, ending 0.5 mm above the tips of the shanks. The fiber was connected with a light source (473 nm, 50 mW) or in some control experiments with 635 nm 250 mW output light source. Light was flashed for 10 ms in every 400 ms or 5 s for at least 200 cycles while the animal was resting. Each recording lasted 30–90 min.

Recordings were performed with Intan RHD-2000 system (RHD Recording Controller 2.01, Intan Technologies, USA). Signals were amplified (200×), filtered (7500 Hz low pass), and digitized at 20 kHz. The animal's behavior was recorded by tracking the ball's movement and by a video camera synchronized with the recording system. Detected spikes were sorted with the help of KlustaKwik 1.4 and Klusters software[52]. Twenty-four-dimensional feature vectors for each spike were generated with principal component analysis. Clusters of feature vectors were manually checked, and only clear clusters were considered as spikes from a single cell. Analysis was performed with custom written codes on Matlab 2018b.

**Y-maze test**. Y-maze testing was carried out in a Y-shape maze with three gray, non-reflective plastic arms (width: 5 cm; length: 35 cm; height: 10 cm). The animals were transferred to the testing room 2 h before experiments. Animals (N = 9, 8-month-old male SOM-Cre-ChR2, N = 5 PV-Cre-ChR2, and N = 4 SOM-Cre-EGFP mice) were placed into the center of the maze and then allowed to alternate freely in the apparatus for 5 min in dim light conditions without any reinforcement. The inner surface of the maze was cleaned with 70% ethanol between each trial and was allowed to dry. An entire entry was counted when all three body points (nose, center, tail base) were inside the arm. The sessions were video-tracked and analyzed with Noldus EthoVision XT (Noldus Information Technology, Netherlands). When animals entered the central zone (10 cm in each arm from center), bilateral MEC photostimulation (50 ms, 5 Hz) with 465 nm light (Doric Lenses) was performed.

Spontaneous alternation was defined as permanent entries into each of the three arms (A, B, C) on overlapping triplet sets (e.g., ABC; BCA; CAB…). The percentage of spontaneous alternation performance (SAP%) was calculated by dividing the number of alternations by the number of possible alternations (total arm entries − 2) ×100. All tests were carried out at the same time of the day.

**Morris water maze**. The spatial learning ability (N = 8, SOM-Cre-ChR2, N = 4 SOM-Cre-EGFP, and N = 5 PV-Cre-ChR2 male 3–6 months mice; N = 4 SOM-Cre-ChR2 and N = 5 PV-Cre ChR2 mice were not involved in the Y-maze experiments, the rest of the animals performed both tests) was tested using the Morris water maze task in dim light conditions. A circular 122 cm diameter white pool was filled with 24 ± 2 °C water. The pool was divided into 4 quadrants and the hidden, transparent platform (d = 10 cm) was submerged 1 cm below the water surface at a fixed position in the southeast quadrant. Some distal cues were placed outside the pool to help spatial navigation. The task comprised a 5-day spatial acquisition phase (4 consecutive trials/day, semi-random start positions) and a 1-day probe trial. Mice were placed in the water facing the tank wall and allowed to acquire for 60 s or until the platform was discovered. Animals were permitted to rest on the platform for 30 s. If animals did not reach the platform after 60 s, they were gently guided to the platform. Twenty-four hours after the last acquisition day, the probe trial was performed. The platform was removed, the animals were placed in the water from a new starting position and they were allowed to swim freely for 60 s while MEC photostimulation (50 ms, 5 Hz) by exciting (465 nm) light was executed. Acquisitions, path-tracking, and escape latency analysis were performed with Noldus EthoVision XT (Noldus Information Technology, Netherlands). The experimenters were blind to the phenotypes of the animals.

**Immunohistochemistry, confocal and STED imaging, electron microscopy**. After 1–2 h of the in vivo electrophysiological experiments, non-recorded control animals were deeply anesthetized and transcardially perfused with ice-cold saline and then with 4% paraformaldehyde and 0.1% glutaraldehyde dissolved in 0.1 M PB (pH = 7.4). Brains and immersion fixed acute slices were sectioned into 60- or 70-μm-thin sections with a vibratome (Leica, VS1200s).

On the selected sections, immunoreactivities were tested: PV (rabbit, 1:30,000, Swant, PV25; mouse, 1:30,000, Sigma-Aldrich, P3088) Reelin (mouse, 1:500, Millipore, MAB5364) SOM (rabbit, 1:3000, Peninsula Laboratories, T-4103) Wfs1 (rabbit, 1:1000, ProteinTech, 11558-AP) and PCP4 (rabbit, 1:250, SantaCruz, sc74816) were diluted in 0.1 M PB and incubated overnight at room temperature. For detection, fluorescent dye (Alexa488/Alexa594/Alexa633) conjugated donkey secondary antibodies (Jackson ImmunoResearch) and Aberrior STAR ORANGE conjugated goat antibodies raised against the host species of primary antibodies were applied on the sections. Confocal images were taken with a Zeiss LSM710 confocal microscope with ×20 and ×63 objectives. STED images were taken with ×100 objective with an Abberior Expert Line STED system assembled on a Nikon inverted microscope; 775 nm laser intensity were set between 7 and 25% of maximal power, pinhole 0.62 AU, pixel size 30 nm. For electron microscopy, sections were incubated at room temperature in rabbit-anti-mCherry (Abcam, 1:30,000, 0.1 M PB). After extensive washes, sections were incubated for 8 h in biotinylated donkey-anti-rabbit (Jackson Immunoresearch, 1:500) solution. Sections were developed with the standard Avidin–Biotin Peroxidase Kit (Vectastain, 1:500) and diamino-benzidine (Sigma-Aldrich) was used as a chromogen. After treatment with 1% Osmium tetroxide (SPI Supplies) and 1% Uranyl acetate (Amersham), slices were embedded in durcupan (Sigma-Aldrich). Sixty-nm ultrathin sections were contrasted with 3% Lead citrate (Leica) and investigated with a Jeol JEM 1400-Plus transmission electron microscope. Digital images were brightness/contrast adjusted with ImageJ.

**Statistics and reproducibility**. Normalities of samples were tested with D'Agostino–Pearson test. Normally distributed samples were compared with T test; non-normally distributed data were compared with Mann–Whitney test. Kruskal–Wallis test followed by post hoc Dunn's pairwise comparison tests for non-normally distributed data and ANOVA test with Tukey multiple comparison test for normally distributed data were used for three or more group comparison. Drug and light effects were compared with paired T test. Data were presented as mean ± S.E.M. No statistical methods were used to predetermine sample sizes, but our sample sizes are similar to those reported in previous publications in the field. The animal and the cell numbers are presented as N and n, respectively.

## Data availability

All source data for the figures are listed in Supplementary Data 1. All relevant data are available from C.V. (csaba.varga@aok.pte.hu).

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

## Acknowledgements

We thank B. Boran and Z. Kraboth for excellent technical assistance. SST4 KO mice and J-2156 SOM receptor antagonist were generously donated by Z. Helyes. We thank G. Kriszta and L. Péczely for advice on behavioral experiments. C.V. acknowledges the support from Hungarian Brain Research Program (20017-1.2.1-NKP-2017-00002), EU Social Fund (EFOP-3.6.2-16-2017-00008), and GINOP 2.3.3-15-2016-00026.

## Author contributions

M.K., N.H.-M., A.A.-L., and C.V. designed the experiments; all the authors performed experiments, analyzed data, and commented on the manuscript; M.K., N.H.-M., and C.V. wrote the manuscript.

## Competing interests

The authors declare no competing interests.
