## [Peer Review File · Communications Biology]

Reviewers' Comments:

Reviewer #1:

Remarks to the Author:

The authors explore the functional features of SOM+GABAergic interneurons in MEC and demonstrated successfully that such dendritic GABA-SOM dual inhibition influences short-term memory formation without affecting spatial navigation. They propose a novel role of SOM as a neuromodulator, representing a significant achievement in the field of neurophysiology. However, I feel that interpretation of morphological findings (things like morphological features of SOM+ interneurons in MEC) are not fully discussed. Anatomical description should be added before I would recommend this manuscript for publication.

Major points

1) The authors generally compare layers III-V with layer II of MEC. Commonly, layers II and III of the neocortex are together referred to as the supragranular layers, which are dedicated to inter- and intra-cortical connections, contrasting with the rich subcortical connectivity of the infragranular layers (layers V-VI). Along with the intent of Figure 2, in which the authors compare the strength of inhibitions on pyramidal cells in supra-, infra-, and granular layers of the somatosensory cortex, comparing layers II-III with layers V-VI in MEC is more appropriate. Alternatively, the authors should clearly explain the rationale of their comparison between layers III-V and layer II. When more appropriate, the term "layers III and V" is better than "layer III-V", unless the authors can state findings for layer IV neurons.

2) The location (layer) and morphology of recorded target neurons should be demonstrated in several figures (especially in Figs. 1, 3b, and 5). The authors should explain why they presume that the inhibited target neurons shown in Fig. 3b and Fig. 5 are pyramidal cells in layers III-V. Inserting photomicrographs or schemes that indicate the locations of soma for target neurons would be preferable. This is all the more necessary because the figures indicate very important findings of a novel inhibitory pattern of SOM+ interneurons in vivo. Also, in Fig. 1, it is uncertain whether the labeled neurons are actually located in layers III and V, because the boundaries among layers are not displayed in the photomicrographs of f and i. For the same reason, again in Fig. 2f, boundaries among the stratum radiatum, pyramidal cell layer, and stratum oriens of CA1 should be represented.

3) Interpretations of morphological findings are not fully discussed. Discussing the morphological and cytoarchitectonical features of interneurons is necessary (J Neurosci 24(12):2853-2865, 2004), because the forms of axonal and dendritic processes of interneurons are clearly presented in Fig. 1c, d. As for Supplementary Fig. 1, inserting descriptions of the features of labeled boutons (for instance, size, shape, and distribution density) and processes (thickness and distribution) with respect to each layer would be preferable.

4) Is it right to think that the soma of observed PV+ and SOM+ interneurons are mostly distributed in layer II of MEC? Assuming that the mouse MEC has patchy modular circuit structures similar to the rat EC (Frontiers in Systems Neuroscience 2017, doi: 10.3389/fnsys.2017.00020, J Neurophysiol 119: 2129-2144, 2018), is it possible to incorporate SOM+ interneurons in such a functional model?

5) The structural definition of MEC is unclear, especially in Fig. 1 and Supplementary Fig. 1. If possible, presenting photomicrographs of Nissl- or NeuN-stained sections would be better, in order to indicate the cytoarchitecture of MEC. The authors described that they observed the dorsal MEC in Supplement Fig. 1, but how were the dorsal and ventral MEC defined?

In Supplement Fig. 1, boundaries between layers III and IV, layers IV and V, and layers V and VI should be inserted. The thickness of layer V seems to be quite wide (over 200 μm ?), if the position of rectangle d is actually within layer V. If the cutting plane of the section is horizontal, this

thickness of layer V would be questionable.

It would be great if a statement could be added detailing why the authors' observations are limited to MEC (not including LEC).

Minor points

1) Fig. 1e-g is not cited in the Results. I think the second paragraph in p.5 will be appropriate, but the findings of Fig. 1h-j (SOM) are described prior to e-g (PV) in the Results and seem somewhat inadequate in view of the order of description.

2, This is a matter of terminology, but I think the authors should clearly define the meaning of the term "principal cells". As in the description explaining Fig. 5a-d in the Results and the Figure legends, "principal cells" seems to be too easy to confuse with "pyramidal cells". Is it correct that the term "principal cells" in this manuscript includes "stellate cells and pyramidal cells in layer II" and "pyramidal cells in layers III-V (or III and V)" ?

3) The cutting plane of the sections shown in Fig. 1, Fig. 2, and Supplementary Fig. 1 should be mentioned in the Figure legends.

4) The word "importantly" is heavily used through the Results and Discussion sections. Replacing at least some of these uses of the term with another expression would be better.

5) p.6, line 15: Replace "hippocampus CA1" with "hippocampal CA1".

6) In the Figure legends for Fig. 1, p.20, the bottom line and p.21, line 4: replace "Chr2" with "ChR2".
p.21, line 9, replace "PV-ChR2 (f) and SOM-ChR2 (i) MEC" with "PV-ChR2 (f) and SOM-ChR2 (i) in MEC"

7) Fig. 6 is rather difficult to understand. It would be nice if the authors could provide the layer number and other labels such as PV and SOM.

Reviewer #2:

Remarks to the Author:

This study uses optogenetic approaches to assess the functional connectivity of parvalbumin expressing (PV+) and somatostatin expressing (Som+) interneurons in the medial entorhinal cortex. The key advances are: 1) Whereas PV+ interneurons innervate principal cells in layer 2-5 of the MEC, Som+ interneurons are selective for principal cells in layers 3-5; 2) The action of Som+ interneurons involves a fast GABA-mediated component and a slower somatostatin-mediated component. Evidence is also provided that Som+ cells selectively influence short-term memory without affecting spatial memory. The circuit analyses will be an important contribution, in particular because Som+ interneurons are thought to be specifically associated with non-grid cells. The behavioural experiments are less convincing.

Major points

1. The composition of the intracellular recording solution is important to interpretation of the data. It appears that more than one solution may have been used although only one is given in the methods.

- It appears that the intracellular solution in Figure 5 may be different to earlier figures (responses

are hyperpolarizing). Please clarify.

- p 6, para 2. Are these experiments also using CsCl intracellular? If so, why are the responses smaller? Either way its important to make this clear.

- p 5, para 1. It would be helpful to let the reader know earlier that experiments are in CsCl. Key here is that the GABA-reversal potential is 0 mV so large depolarizing PSPs are expected. Potential cause of small responses in physiological GABA is not poor space clamp but small driving force.

- In the Methods please clearly state the Cl⁻ equilibrium potential given the solutions used.

2. Figure 1. Pharmacology to show responses are GABA-ergic would be helpful here.

3. Figures 1 and 3. As well as showing examples and averages of responses it is important here to show summary plots that indicate the distribution across the population of response amplitudes and kinetics.

4. I don't find the behavioural experiments to be all that convincing. I suggest either to carry out additional controls suggested below, or to make the conclusions extremely tentative and to clearly discuss the limitations.

- Are the animals used for Water-maze experiments the same as used for T maze experiments? If not, how can the possibility that the neurons were insufficiently activated be ruled out? Some sort of a positive control for the watermaze experiment is important here.

- Perhaps Som⁺ neurons are required for acquisition rather than recall of water-maze memories? Can this be tested? If not, this possibility should be discussed. In this case, it's not correct to conclude that Som⁺ neurons do not influence spatial memory, only that they don't seem to be involved in its recall.

- The watermaze experiment has too few mice to demonstrate a null result (n = 4 ChR2 mice and n = 3 controls).

- For the Y-maze experiments an EGFP control would be more convincing than a different light wavelength.

5. p 9. Are the n values for cells or animals (e.g. n = 64, 81). To avoid pseudoreplication this should be animals, or the analysis should be nested.

6. Figure 5d, f and 6e should show the distribution of the data.

7. Which part of the brain is data in Figure 3 from? The Methods indicate that recordings were made from hippocampus and MEC.

8. The methods should indicate how groups sizes were determined and whether blinding was carried out for experiments that compare multiple groups.

** Minor

p 5, para 1. "1.9 ±0.12 ms and 2 ±0.17 ms respectively". Presumably this is referring to the latency of the responses?

p 6, para 1. "all layers of the MEC". Better to stick to L1-5. The experiments don't investigate layer 6.

p 11, para 1. "Optogenetically tagged PV+ interneurons in the MEC do not show spatial selectivity 14". This is incorrect. This study shows that PV+ interneurons encode spatial information but they don't have classic grid fields.

Reviewer #3:

Remarks to the Author:

The manuscript entitled "Coaction of somatostatin and GABA modulates working memory formation in the medial entorhinal cortex" by Miklós Kecskés, Nóra Henn-Mike, Ágnes Agócs-Laboda et al. presents results from a study on mice, investigating the physiology of somatostatin-positive GABAergic neurons in the medial entorhinal cortex, from their synaptic function to their role in memory formation.

Expressing ChR2 in specific interneurons and doing slice physiology, the authors described the inhibition of somatostatin+ (SOM+) interneurons throughout medial entorhinal cortex (MEC) layers, and found a higher SOM+ inhibitory effect in layers deeper than layer 2. In contrast, parvalbumin+ (PV+) inhibition was homogeneous across MEC layers. They suggest that the SOM+ inhibitory pattern described in MEC is specific to medial entorhinal cortex by comparing to somatosensory cortex S1 and dorsal CA1. Then, they showed, by recording multi-unit activity and activating interneurons via optogenetics in awake head fixed mice, that the inhibition by somatostatin interneurons lasted longer than that of PV interneurons. They noticed that SOM+ inhibition lasted unusually long and hypothesized that it could actually be due to somatostatin itself, known to be a modulator of synaptic activity in other areas. Using ultrastructure imaging, they checked that somatostatin was actually present in the presynaptic compartment of symmetrical synapses, which suggests a potential co-release of somatostatin with GABA while SOM+ cells are activated. They did not report longer lasting inhibitory GABA events in layer 3-5 cells triggered by optogenetic activation of SOM+ cells, compared to optogenetic activation of PV+ cells. GABAB receptors did not seem to be involved in SOM+ cell postsynaptic inhibition. To test whether somatostatin was involved in the SOM+ evoked longer lasting inhibition, they crossed a STTR4 (somatostatin receptor type 4) KO mouse with their ChR2 SOM+ mouse, and performed optogenetics experiments in vitro. Using patch clamp in slices and multi-unit recordings in awake head fixed mice, they saw that SOM+ inhibition in SOM-Chr2/STTr4 KO mice was shorter than in the SOM-Chr2 mice, suggesting a role for somatostatin in the longer lasting inhibition by SOM+ cells. Additionally, they show that somatostatin agonist tonically reduced layer 3-5 cells firing in vitro. Finally, they showed that an overactivation of SOM+ cells in MEC caused disturbance in a working memory task, but did not impair long term memory.

I think that the authors put together an impressive set of diverse and complementary experiments, in order to understand somatostatin+ cell function in neural circuits underlying memory processes. The findings presented by the manuscript are original and present an immediate relevance for a broad audience in neuroscience, not only in the field of learning and memory circuits but also for research on cortical microcircuit processing and modeling as well as for interneurons' aficionados. Somatostatin is certainly one of the most known molecular markers commonly used to identify dendrite targeting interneurons in cortical areas (such as MEC in the current study). It is a neuropeptide, which as the authors show in the present study, can be

released as a neurotransmitter, but its role in the physiology of neuronal circuits or impact of behavior has been surprisingly overlooked. The current study may show only one specific role (but yet important) of somatostatin, but it certainly demonstrates that more effort should be done in that direction if we want to understand brain circuits.

I highly support the publication of this work, however, I believe that a significant revision effort is required before being suitable for publication in *Communication Biology*.

I was mostly disappointed because of the title: "Coaction of somatostatin and GABA modulates working memory formation in the medial entorhinal cortex". I was expecting to see a specific role for somatostatin during behavior, but the experiments carried out do not really show it. In my opinion, this is the main weakness of the paper. It seems that the authors should be able to address that problem, based on the experiments they have already presented in the paper.

Including this one, my comments are listed below:

1/ I feel like the role of somatostatin vs GABA on memory has been slightly overlooked, both in results and the discussion. GABA and Somatostatin inhibitory roles have been shown by the experiments conducted in vivo and in vitro, but their individual roles and coaction during behavior remains unresolved.

In figure 5, the authors used a SSTR4 KO mouse to show that inhibition by SOM+ cells not only depends on GABAA, but also on somatostatin release. I think the results are quite clear. I am wondering why the authors did not use the similar mouse line for their behavioral experiment (figure 6). They report an effect of SOM+ cell activation on working memory and long term memory, but whether this role involves the "coaction" of GABA and somatostatin is not addressed. -Using the SSTR4 KO / SOM+Chr2 and following the same paradigm could bring some answers to that question. I think, it would at least address whether the impairment of working memory by overactivation of SOM+ cells is caused by co-release of GABA and somatostatin, or if only GABA release would lead to such an impairment. There might be a limit that I am unaware of here. The SSTR4 KO might have a strong phenotype concerning working memory, or long term memory, or worse. However, if the test can be run with these mice, it could still be used to try to answer that question.

-Additionally/alternatively, testing whether somatostatin receptor activation alone would disturb working memory would be of great interest. Injection of somatostatin agonists into MEC could be performed prior to the task to test if it could, alone, disrupt working memory.

-The authors should show how the stimulation activate SOM+ cells, and inhibit other cells (similar experiments as in Fig 3 or 5, but the type of protocol used during behavior).

I am aware that this requires substantial efforts, but I believe that the added value would be highly significant, no matter what they would show.

-In the discussion, the limitation of the behavioral experiment and the potential role of somatostatin is not very developed. What about MEC L3-5 SOM+ cells activity during the behavioral tasks described here? Disrupting their activity clearly impairs performance, but what is their role? When do they fire? I am not asking the authors to do the experiments to answer those questions, the data might be available in the literature. In any case this should be specifically discussed, and future direction should be suggested.

2/ The results and conclusions of the manuscript are not based on proper statistical analyses, and a proper statistics section is missing in the methods.

I noted several points where statistic were missing, or inappropriately used. I encourage the authors to read and apply the guideline from communications biology (<https://www.nature.com/commsbio/submit/submission-guidelines#statistical-guidelines>) and perhaps to confirm their statistical approach with colleagues. Maybe they could make a list with the details of all the statistical tests they did in a separate file (especially for a subsequent submission).

Here are some mistakes I noticed:

◇ There is no statistical comparison of data relating to Figure 1 and 2. The results seem very clear, and I doubt that any statistic is going to prove you wrong, but they should still be done.

Page 7 "(50% recovery from inhibition: PV-Cre-ChR2: 37 ± 1 ms $n=136$, SOM-Cre-ChR2: 89 ± 7 ms $n=51$, $p < 0.0001$, Wilcoxon rank sum test, Fig. 3b and Fig. 5f)." ◇ It seems that the Wilcoxon rank sum test relates to figure 5 only (but figure 3 and 5 have similar panels, so it is a bit confusing – also the test is not appropriate, see below)

Page 8 – "In our experiments, however, neither rise times (SOM: 5.96 ± 1.22 ms ($n=11$), PV: 5.77 ± 0.99 ms ($n=11$)), nor decay times (SOM: 179 ± 34.5 ms, PV: 199.8 ± 21.6 ms) differed between the two groups (Fig. 5a)" ◇ The statistical comparison is missing.

Page 9 – "This agonist had no effect on the firing frequency of layer II principal cells (19.8 ± 1.9 Hz control vs 18.4 ± 2.4 Hz J-2156 @300 pA injected current, $n=9$, Fig. 5g,h), but decreased the firing frequency in layer III-V pyramidal cells (15.1 ± 1.2 Hz control vs 11.1 ± 1.3 Hz J-2156 @150 pA, $n=13$, Fig. 5g,h)". ◇ The statistics used here are not appropriate. The differences shown on the figure seem quite big, and I have no doubt that it will remain significant, but my point here is to use proper statistics. They authors use the "Wilcoxon rank sum test" 2 times (comparing SOM+ activation and PV+ activation, and SOM+ activation with SOM+ activation in Sstr4 KO). I believe this test is equivalent to a Mann-Whitney U-test, which is not appropriate when there are more than two groups of independent observations. At least the authors should correct p-values.

3/ All along the manuscript, cells from L3 and L5 have been grouped together. However L3 and L5 in MEC are anatomically distinct, have different cell types or clearly distinct input/output connectivity patterns. Why were they grouped together? If they were because the response to inhibitory cells were similar, or because they are just considered as another group from L2 cells, then the authors should say so, and most importantly, they should justify it. Maybe the amplitude of optogenetically-triggered SOM+ IPSPs or the effect of SOM agonists are different between L3 and L5?

4/ The authors wrote "Layer II pyramidal and stellate cells are the closest located to layer I, where the most somatostatin immunoreactive axonal cloud can be found (Supplementary Fig. 1)" (page 5)

On Supplementary Fig. 1, I don't see the difference of somatostatin expression between layer 1 and 2, neither in a nor b. I don't deny that it exists, but the figure does not show it. Maybe the authors could consider showing a non-overlaid version of a. Adding a simple quantification of the background fluorescent signal along the radial axis and aligned with the figure could help. The authors mention the repartition of SOM fluorescent signal in EC, but not S1 and CA1. I am curious, is there anything in the fluorescent signal repartition that could have predicted the difference of responses between layers in EC and the similarity of the responses in S1 and CA1?

5/ The results part should be improved for clarity. The authors should check that they clearly mention the methods that they used and that the figures are ordered optimally.

Here are listed a few things/suggestions:

The sequence Fig 3 Fig 4 Fig 5, and corresponding text could be revised. Figure 5d contains similar results than figure 3. Maybe Fig 3 and 5 should be grouped together and figure 4 could alternatively come after or as a supplementary.

I noticed that the text describes the results of stimulation of SOM cells before PV cells whereas figure 1 order is the opposite. Reordering the text and the figure panels consistently would make sense. It seems that the authors change the order of the text given that more details about the methods are given in the second paragraph about PV. If the text is kept that way, these method details should be moved to the SOM paragraph.

Page 6 " Here, we utilized the general somatostatin promoter driven Cre mouse line (SOM-Cre) to express ChR2 in all types of SOM cells in both somatosensory cortex and dorsal hippocampus CA1" It seems to be the same technique as used earlier, but it is not clear how ChR2 is brought here (breeding mouse lines, injection of AAV).

More generally, it is sometimes not clear whether the expression of ChR2 is obtained after breeding or viral injections (both are described in the methods, but what is the approach should be

clearly stated in the results and figures when mentioned).

6/ Can the authors briefly justify their use of SST-R4 KO mouse? There are 5 types of SST receptors, I am just wondering why this mouse in particular. Is it present in MEC L3-5? Is it more specific to MEC than other receptors? Easy access to that particular line would also be a legit justification as long as SST-R4 is expressed in all layers of MEC.

7/ The role of somatostatin on the modulation of neuronal activity is only mentioned in the result section, but it should be described in the introduction.

8/ The main results should be briefly summarized at the beginning of the discussion.

9/ The discussion sometimes reads more like an introduction. There are a lot of references to the literature (which is of course essential), but maybe too much (some parts are actually repeated from the introduction). My point is that it is interesting, but I feel like some of the current paper's results are a bit lost in it, rather than being emphasized.

10/ Page 5. "poor space-clamping (intracellular solution contains CsCl, see methods)"
Cs solution limits space clamp, but it does not prevent it at all, especially in distal dendrites (<http://www.nature.com/articles/nn.2137>). Please, remove that statement, and discuss space clamp as a potential but minor limitation of that experiment (I think that it could be done in the methods as a justification for using cs-based internal).

11/ The effect was monosynaptic on both stellate and pyramidal cells (1.9 ± 0.12 ms and 2 ± 0.17 ms respectively) ; I would use "latency of events" rather than "effect".

12/ The latency of PV+ and SOM+ evoked IPSPs are mentioned for Layer 2 cells but not for 3-5 cells. Also it was not mentioned for CA1 and S1 cells I believe.

13/ Figure 5c. Even though the indication of the light pulse could be used as a time scale, a proper time scale bar is missing.

14/ The authors should briefly describe, in the result section, what protocol they use for light delivery during the behavioral experiment and, ideally, they. It is not clear in the methods for how long the light stimulations (bilateral MEC, 50ms, 5Hz) were delivered (for both Y maze and Morris WM).

15/ I thought that the figures were nice and clear.

Just for Figure 5d, 5f, or 6e, I am not a big fan of the bar-graph representation, I think showing all the data points and the median (or mean depending on the statistical test you choose), as in figure 2, would be much nicer. Box/whisker plots are also an alternative. Also in 5h (and 5d), please indicate what the error bars represent.

Reviewers' comments:

Reviewer #1 (Remarks to the Author):

The authors explore the functional features of SOM+GABAergic interneurons in MEC and demonstrated successfully that such dendritic GABA-SOM dual inhibition influences short-term memory formation without affecting spatial navigation. They propose a novel role of SOM as a neuromodulator, representing a significant achievement in the field of neurophysiology. However, I feel that interpretation of morphological findings (things like morphological features of SOM+ interneurons in MEC) are not fully discussed. Anatomical description should be added before I would recommend this manuscript for publication.

We thank the reviewer for the suggestions and comments that helped to strengthen the anatomical description of the study.

Major points

1) The authors generally compare layers III-V with layer II of MEC. Commonly, layers II and III of the neocortex are together referred to as the supragranular layers, which are dedicated to inter- and intra-cortical connections, contrasting with the rich subcortical connectivity of the infragranular layers (layers V-VI). Along with the intent of Figure 2, in which the authors compare the strength of inhibitions on pyramidal cells in supra-, infra-, and granular layers of the somatosensory cortex, comparing layers II-III with layers V-VI in MEC is more appropriate. Alternatively, the authors should clearly explain the rationale of their comparison between layers III-V and layer II. When more appropriate, the term “layers III and V” is better than “layer III-V”, unless the authors can state findings for layer IV neurons.

Our response:

Indeed, in the neocortex layerII and layerIII pyramidal cells are referred together, due to their largely similar connectivity and morphology. However, in the MEC, layerII and layerIII have less in common. LayerII contains mainly stellate cells which project to the dentate gyrus, meanwhile layerIII pyramidal cells project to the CA1-CA3 stratum lacunosum moleculare. Their connectivity with local neurons also differ. Moreover, there is a striking difference in the information they code: layerIII pyramidal cells have been shown to be crucial in temporal memory formation and layerII codes mostly spatial information (grid cells are mostly located in layerII). Also, their electrophysiological properties differ largely: layerIII pyramidal cells (and layerV cells as well) are capable of persistent firing (believed to play a major role in memory formation processes), meanwhile layerII cells are known to show subthreshold membrane oscillations, which might be important in grid cell formation.

LayerIV -also called as lamina dissecans- do not contain principal cells, which is a unique feature of the entorhinal cortex. Therefore, in line with other studies, we did not investigate and

discussed this layer in our manuscript. We added a short description in the results regarding layerIV and also about why we have not recorded layerVI pyramidal cells.

We explain the rationale of comparing layerII to deeper layers in the Results section in the current version more thoroughly.

2) The location (layer) and morphology of recorded target neurons should be demonstrated in several figures (especially in Figs. 1, 3b, and 5). The authors should explain why they presume that the inhibited target neurons shown in Fig. 3b and Fig. 5 are pyramidal cells in layers III-V. Inserting photomicrographs or schemes that indicate the locations of soma for target neurons would be preferable. This is all the more necessary because the figures indicate very important findings of a novel inhibitory pattern of SOM+ interneurons in vivo. Also, in Fig. 1, it is uncertain whether the labeled neurons are actually located in layers III and V, because the boundaries among layers are not displayed in the photomicrographs of f and i. For the same reason, again in Fig. 2f, boundaries among the stratum radiatum, pyramidal cell layer, and stratum oriens of CA1 should be represented.

Our response: We have put labels showing the boundaries among layers on the figures in order to show more clearly where the biocytin filled cells are located. In the in vivo experiments the cells are not labeled, therefore we cannot show their exact location. Moreover, there is no published electrophysiological feature which would enable us to distinguish between layerIII and layerV pyramidal cells in the MEC. We relied on the silicon probe tracks, and on the fact that pyramidal cells have wider action potentials than interneurons. We have put examples of extracellular action potentials on Figure 3 to represent the shape of action potentials of pyramidal cells and interneurons.

3) Interpretations of morphological findings are not fully discussed. Discussing the morphological and cytoarchitectonical features of interneurons is necessary (J Neurosci 24(12):2853-2865, 2004), because the forms of axonal and dendritic processes of interneurons are clearly presented in Fig. 1c, d. As for Supplementary Fig. 1, inserting descriptions of the features of labeled boutons (for instance, size, shape, and distribution density) and processes (thickness and distribution) with respect to each layer would be preferable.

Our response: We added detailed morphological analysis of the distribution and the targets of SOM/mCherry expressing neurons in all layers of the MEC. We analysed data from 5 animals and over 150 boutons. This data is inserted into the text and electronmicroscopical images have been added to Supplementary Figure 1.

Previous publications have shown that the dendritic morphology of interneurons is not a distinctive feature of GABAergic interneurons (see for example: Maccaferri, 2005, Stratum oriens interneuron diversity and hippocampal network dynamics, J. Physiol.; Pawelzik et al. 2002, Physiological and morphological diversity of immunocytochemically defined parvalbuni- and cholecystocikin-positive interneurons in CA1 of the adult rat hippocampus, Journal of Comparative Neurology). Rather, the axonal targets specify the different interneuronal cell

types. Dendritic morphology predicts the type of converging inputs on a given cell population. The PV+ and SOM+ interneurons' innervation by local networks or by inputs arriving from outside of MEC is beyond the scope of the present study. However, we have added new images in Supplementary Figure 1, where the heterogeneous dendritic morphology of both cell populations can be seen.

4) Is it right to think that the soma of observed PV+ and SOM+ interneurons are mostly distributed in layer II of MEC? Assuming that the mouse MEC has patchy modular circuit structures similar to the rat EC (Frontiers in Systems Neuroscience 2017, doi: 10.3389/fnsys.2017.00020, J Neurophysiol 119: 2129-2144, 2018), is it possible to incorporate SOM+ interneurons in such a functional model?

Our response: Previous publications focused mostly on circuits of layerII and recorded principal cell-interneuron connectivity in this layer. Here we have extended the focus to multiple layers and shown that both PV+ and SOM+ interneurons are abundant in layerII-V. We suggest in the discussion that the weaker dendritic inhibition by the SOM+ cells on layerII cells might play a role in for example the generation of grid-cell firing. The patchiness of layerII pyramidal cells' distribution has not been proven to hold functionality: both pyramidal cells and stellate cells (homogeneously distributed cell population in layerII) have been proven to show grid-cell firing. It is currently unknown whether layerII cell islands (patches) code similar information or innervate each other heavily or receive similar converging information from structures outside of the MEC.

5) The structural definition of MEC is unclear, especially in Fig. 1 and Supplementary Fig. 1. If possible, presenting photomicrographs of Nissl- or NeuN-stained sections would be better, in order to indicate the cytoarchitecture of MEC. The authors described that they observed the dorsal MEC in Supplement Fig. 1, but how were the dorsal and ventral MEC defined?

In Supplement Fig. 1, boundaries between layers III and IV, layers IV and V, and layers V and VI should be inserted. The thickness of layer V seems to be quite wide (over 200 μm ?), if the position of rectangle d is actually within layer V. If the cutting plane of the section is horizontal, this thickness of layer V would be questionable.

Our response: Done.

We have completely replaced Supplementary Fig 1, and a MEC specific marker is present (WFS1+ cell island in layerII). Moreover, we have labeled the hippocampus/MEC/LEC for better visibility on Fig. 1. Nissl- or NeuN-stainings have not been performed in our experiments.

It would be great if a statement could be added detailing why the authors' observations are limited to MEC (not including LEC).

Our response: Done.

We have added a sentence in the Discussion about the possible similar target selectivity in the LEC.

Minor points

1) Fig. 1e-g is not cited in the Results. I think the second paragraph in p.5 will be appropriate, but the findings of Fig. 1h-j (SOM) are described prior to e-g (PV) in the Results and seem somewhat inadequate in view of the order of description.

Our response: Done.

We have changed Figure 1. We switched the two columns and put the SOM-ChR2 results on the left side, and PV-ChR2 results on the right side and changed the figure labelling in the text accordingly. We have cited all figures in the Results section in the current version.

2, This is a matter of terminology, but I think the authors should clearly define the meaning of the term “principal cells”. As in the description explaining Fig. 5a-d in the Results and the Figure legends, “principal cells” seems to be too easy to confuse with “pyramidal cells”. Is it correct that the term “principal cells” in this manuscript includes “stellate cells and pyramidal cells in layer II” and “pyramidal cells in layers III-V (or III and V)” ?

Our response: Done.

In the current version we added a sentence describing what we consider as principal cells. The term “principal cells” involve stellate and pyramidal cells. We used this expression when we wanted to refer to both cell types (stellate and pyramidal) in the text.

3) The cutting plane of the sections shown in Fig. 1, Fig. 2, and Supplementary Fig. 1 should be mentioned in the Figure legends.

Our response:

Done.

4) The word “importantly” is heavily used through the Results and Discussion sections. Replacing at least some of these uses of the term with another expression would be better.

Our response: Done.

We have completely re-written the Discussion and tried to avoid this word.

5) p.6, line 15: Replace “hippocampus CA1” with “hippocampal CA1”.

Our response: Done.

6) In the Figure legends for Fig. 1,

p.20, the bottom line and p.21, line 4: replace “Chr2” with “ChR2”.

p.21, line 9, replace “PV-ChR2 (f) and SOM-ChR2 (i) MEC” with “PV-ChR2 (f) and SOM-ChR2 (i) in MEC”

Our response: Done.

7) Fig. 6 is rather difficult to understand. It would be nice if the authors could provide the layer number and other labels such as PV and SOM.

Our response: Done.

We added data on PV-ChR2 and SOM-EGFP animals and re-organized Fig 6.

Reviewer #2 (Remarks to the Author):

This study uses optogenetic approaches to assess the functional connectivity of parvalbumin expressing (PV+) and somatostatin expressing (Som+) interneurons in the medial entorhinal cortex. The key advances are: 1) Whereas PV+ interneurons innervate principal cells in layer 2-5 of the MEC, Som+ interneurons are selective for principal cells in layers 3-5; 2) The action of Som+ interneurons involves a fast GABA-mediated component and a slower somatostatin-mediated component. Evidence is also provided that Som+ cells selectively influence short-term memory without affecting spatial memory. The circuit analyses will be an important contribution, in particular because Som+ interneurons are thought to be specifically associated with non-grid cells. The behavioural experiments are less convincing.

We thank the reviewer for the positive evaluation of our manuscript and the valuable suggestions for control experiments and data re-analysis. They substantially helped to strengthen the conclusions of the present study.

Major points

1. The composition of the intracellular recording solution is important to interpretation of the data. It appears that more than one solution may have been used although only one is given in the methods.

- It appears that the intracellular solution in Figure 5 may be different to earlier figures (responses are hyperpolarizing). Please clarify.

- p 6, para 2. Are these experiments also using CsCl intracellular? If so, why are the responses smaller? Either way its important to make this clear.

- p 5, para 1. It would be helpful to let the reader know earlier that experiments are in CsCl. Key here is that the GABA-reversal potential is 0 mV so large depolarizing PSPs are expected. Potential cause of small responses in physiological GABA is not poor space clamp but small driving force.

- In the Methods please clearly state the Cl⁻ equilibrium potential given the solutions used.

Our response: Done.

Indeed, we used 2 different intracellular solutions. One contains CsCl which sets the reversal potential of GABA_A currents at around -27mV. The other intracell solution contains no CsCl and low concentration of Cl ions, resulting in a -78mV reversal potential. We have inserted the missing description to the methods. Also, we made more clear during the results why this high Cl⁻ containing and why normal Cl⁻ containing intracell were used in different experiments.

2. Figure 1. Pharmacology to show responses are GABA-ergic would be helpful here.

Our response: Done.

We have added an entire set of experiments to Supplementary Fig 1. We show here the dynamics of Gabazine action on PV-ChR2 and SOM-ChR2 effects on layerIII pyramidal cells. This experiment also shows how fast somatic Gabazine puff eliminates the light effect in PV-ChR2 animals and SOM-ChR2 animals. With this experiment we have shown that 1, the effect is GABAA receptor driven, and 2, the SOM+ neurons innervate the pyramidal cells more distally. The more distal innervation by SOM+ axons is also supported by the electronmicroscopical investigations of of SOM-ChR2+ and PV+ boutons in layerI-V.

3. Figures 1 and 3. As well as showing examples and averages of responses it is important here to show summary plots that indicate the distribution across the population of response amplitudes and kinetics.

Our response: Done

We have included plots of the data in the figures.

4. I don't find the behavioural experiments to be all that convincing. I suggest either to carry out additional controls suggested below, or to make the conclusions extremely tentative and to clearly discuss the limitations.

- Are the animals used for Water-maze experiments the same as used for T maze experiments? If not, how can the possibility that the neurons were insufficiently activated be ruled out? Some sort of a positive control for the watermaze experiment is important here.

- Perhaps Som+ neurons are required for acquisition rather than recall of water-maze memories? Can this be tested? If not, this possibility should be discussed. In this case, it's not correct to conclude that Som+ neurons do not influence spatial memory, only that they don't seem to be involved in its recall.

- The watermaze experiment has too few mice to demonstrate a null result (n = 4 ChR2 mice and n = 3 controls).

- For the Y-maze experiments an EGFP control would be more convincing than a different light wavelength.

Our response: Done.

We have performed additional behavioral experiments with PV-ChR2 and SOM-EGFP animals (all expressions induced with local injection of AAV vector carrying Cre dependent ChR2 or EGFP). As shown earlier, PV-ChR2 (and also SOM-EGFP) animals do not show impairment of Y-maze alternation while light is delivered into their MEC. Only SOM-ChR2 animals showed lower SAP rates in this test. With the water maze experiments we have shown that none of the investigated groups (PV-ChR2, SOM-ChR2, SOM-EGFP) have impaired spatial navigation/memory when light-activation protocol was performed during finding the location of

the hidden platform. This finding is in agreement with previous reports showing that MEC ablated animals are able to navigate and find the learned locations. Indeed, our experiments do not provide any information about spatial memory formation, because the light delivery experiments were performed on the last day, when the animals have already learned the location of the hidden platform in the water-maze. However, it proves that the spatial navigation skills and the retrieval of learned locations is unchanged. We have changed the conclusions of the behavioral experiments and discussed accordingly.

5. p 9. Are the n values for cells or animals (e.g. n = 64, 81). To avoid pseudoreplication this should be animals, or the analysis should be nested.

Our response: Done.

We specified both animal number (N) and cell number (n) as well in the current version. With in vivo electrophysiology experiments it is commonly accepted that several data points (recorded cells) are included from the same animal. However, we performed comparisons between data points gained in different animals and found no differences between the animals.

6. Figure 5d, f and 6e should show the distribution of the data.

Our response: Done.

We have included proper graphs to show the distributions of the data.

7. Which part of the brain is data in Figure 3 from? The Methods indicate that recordings were made from hippocampus and MEC.

Our response: We recorded LFPs in the hippocampus and units in the MEC. We have checked the silicon probe locations during the sectioning of the brains, and in some experiments, we have also labelled the tracks of the silicon probes.

8. The methods should indicate how groups sizes were determined and whether blinding was carried out for experiments that compare multiple groups.

Our response: Done.

** Minor

p 5, para 1. "1.9 \pm 0.12 ms and 2 \pm 0.17 ms respectively". Presumably this is referring to the latency of the responses?

Our response: Done.

We have clarified that sentence.

p 6, para 1. "all layers of the MEC". Better to stick to L1-5. The experiments don't investigate layer 6.

Our response: Done.

p 11, para 1. "Optogenetically tagged PV+ interneurons in the MEC do not show spatial selectivity 14". This is incorrect. This study shows that PV+ interneurons encode spatial information but they don't have classic grid fields.

Our response: Done.

We have removed this statement and indicated that PV+ interneurons encode moderate spatial information.

Reviewer #3 (Remarks to the Author):

The manuscript entitled "Coaction of somatostatin and GABA modulates working memory formation in the medial entorhinal cortex" by Miklós Kecskés, Nóra Henn-Mike, Ágnes Agócs-Laboda et al. presents results from a study on mice, investigating the physiology of somatostatin-positive GABAergic neurons in the medial entorhinal cortex, from their synaptic function to their role in memory formation.

Expressing ChR2 in specific interneurons and doing slice physiology, the authors described the inhibition of somatostatin+ (SOM+) interneurons throughout medial entorhinal cortex (MEC) layers, and found a higher SOM+ inhibitory effect in layers deeper than layer 2. In contrast, parvalbumin+ (PV+) inhibition was homogeneous across MEC layers. They suggest that the SOM+ inhibitory pattern described in MEC is specific to medial entorhinal cortex by comparing to somatosensory cortex S1 and dorsal CA1. Then, they showed, by recording multi-unit activity and activating interneurons via optogenetics in awake head fixed mice, that the inhibition by somatostatin interneurons lasted longer than that of PV interneurons. They noticed that SOM+ inhibition lasted unusually long and hypothesized that it could actually be due to somatostatin itself, known to be a modulator of synaptic activity in other areas. Using ultrastructure imaging, they checked that somatostatin was actually present in the presynaptic compartment of symmetrical synapses, which suggests a potential co-release of somatostatin with GABA while SOM+ cells are activated. They did not report longer lasting inhibitory GABA events in layer 3-5 cells triggered by optogenetic activation of SOM+ cells, compared to optogenetic activation of PV+ cells. GABAB receptors did not seem to be involved in SOM+ cell postsynaptic inhibition. To test whether somatostatin was involved in the SOM+ evoked longer lasting inhibition, they crossed a STTR4 (somatostatin receptor type 4) KO mouse with their ChR2 SOM+ mouse, and performed optogenetics experiments in vitro. Using patch clamp in slices and multi-unit recordings in awake head fixed mice, they saw that SOM+ inhibition in SOM-Chr2/STTr4 KO mice was shorter than in the SOM-Chr2 mice, suggesting a role for somatostatin in the longer lasting inhibition by SOM+ cells. Additionally, they show that somatostatin agonist tonically reduced layer 3-5 cells firing in vitro.

Finally, they showed that an overactivation of SOM+ cells in MEC caused disturbance in a working memory task, but did not impair long term memory.

I think that the authors put together an impressive set of diverse and complementary experiments, in order to understand somatostatin+ cell function in neural circuits underlying

memory processes. The findings presented by the manuscript are original and present an immediate relevance for a broad audience in neuroscience, not only in the field of learning and memory circuits but also for research on cortical microcircuit processing and modeling as well as for interneurons' aficionados. Somatostatin is certainly one of the most known molecular marker commonly used to identify dendrite targeting interneurons in cortical areas (such as MEC in the current study). It is a neuropeptide, which as the authors show in the present study, can be released as a neurotransmitter, but its role in the physiology of neuronal circuits or impact of behavior has been surprisingly overlooked. The current study may show only one specific role (but yet important) of somatostatin, but it certainly demonstrates that more effort should be done in that direction if we want to understand brain circuits.

I highly support the publication of this work, however, I believe that a significant revision effort is required before being suitable for publication in *Communication Biology*.

I was mostly disappointed because of the title: "Coaction of somatostatin and GABA modulates working memory formation in the medial entorhinal cortex". I was expecting to see a specific role for somatostatin during behavior, but the experiments carried out do not really show it. In my opinion, this is the main weakness of the paper. It seems that the authors should be able to address that problem, based on the experiments they have already presented in the paper.

We thank the reviewer for the support, constructive critiques and fair evaluation of our study.

Including this one, my comments are listed below:

1/ I feel like the role of somatostatin vs GABA on memory has been slightly overlooked, both in results and the discussion. GABA and Somatostatin inhibitory roles have been shown by the experiments conducted in vivo and in vitro, but their individual roles and coaction during behavior remains unresolved.

In figure 5, the authors used a SSTR4 KO mouse to show that inhibition by SOM+ cells not only depends on GABA, but also on somatostatin release. I think the results are quite clear. I am wondering why the authors did not use the similar mouse line for their behavioral experiment (figure 6). They report an effect of SOM+ cell activation on working memory and long term memory, but whether this role involves the "coaction" of GABA and somatostatin is not addressed.

-Using the SSTR4 KO / SOM+ChR2 and following the same paradigm could bring some answers to that question. I think, it would at least address whether the impairment of working memory by overactivation of SOM+ cells is caused by co-release of GABA and somatostatin, or if only GABA release would lead to such an impairment. There might be a limit that I am unaware of here. The SSTR4 KO might have a strong phenotype concerning working memory, or long term memory, or worse. However, if the test can be run with these mice, it could still be used to try to answer that question.

-Additionally/alternatively, testing whether somatostatin receptor activation alone would disturb working memory would be of great interest. Injection of somatostatin agonists into MEC could be performed prior to the task to test if it could, alone, disrupt working memory.

-The authors should show how the stimulation activate SOM+ cells, and inhibit other cells (similar experiments as in Fig 3 or 5, but the type of protocol used during behavior).

I am aware that this requires substantial efforts, but I believe that the added value would be highly significant, no matter what they would show.

-In the discussion, the limitation of the behavioral experiment and the potential role of somatostatin is not very developed. What about MEC L3-5 SOM+ cells activity during the behavioral tasks described here? Disrupting their activity clearly impairs performance, but what is their role? When do they fire? I am not asking the authors to do the experiments to answer those questions, the data might be available in the literature. In any case this should be specifically discussed, and future direction should be suggested.

Our response: The SST4KO we used is not MEC specific and shows several anxiety and depression phenotypes, which might be due to the high SST4 expression level in the amygdala. With these phenotypes the results of any behavioural experiments would be questionable. We discuss this issue now in the manuscript. Applying drugs like SOM agonists systematically or in the MEC would require additional canule implantation/deep anesthesia before the behavior experiments. Anesthesia before any behaviour experiments severely influences the outcome of the experiment. The canule implantation and drug application would largely destroy the MEC and the localization of the drug solely to MEC is also almost impossible.

During the behavior experiments we have not recorded the cells' activity, we have only delivered light. Therefore, we cannot show how the excited and the inhibited cells behaved during and in between light delivery. The behavior of individual pyramidal cells during decision making have not been described. Even their theta modulation is not clarified: some authors reported strong theta modulation, other no modulation. However, it has been shown that if their activity is blocked, the synchronization of high-gamma oscillation between CA1 and MEC will disappear and the animals short term memory will be compromised.

2/ The results and conclusions of the manuscript are not based on proper statistical analyses, and a proper statistics section is missing in the methods.

I noted several points where statistic were missing, or inappropriately used. I encourage the authors to read and apply the guideline from communications biology (<https://www.nature.com/commsbio/submit/submission-guidelines#statistical-guidelines>) and perhaps to confirm their statistical approach with colleagues. Maybe they could make a list with the details of all the statistical tests they did in a separate file (especially for a subsequent submission).

Here are some mistakes I noticed:

There is no statistical comparison of data relating to Figure 1 and 2. The results seems very clear, and I doubt that any statistic is going to prove you wrong, but they should still be done.

Page 7 "(50% recovery from inhibition: PV-Cre-ChR2: 37 ± 1 ms $n=136$, SOM-Cre-ChR2: 89 ± 7 ms $n=51$, $p<0.0001$, Wilcoxon rank sum test, Fig. 3b and Fig. 5f)." It seems that the Wilcoxon

rank sum test relates to figure 5 only (but figure 3 and 5 have similar panels, so it is a bit confusing – also the test is not appropriate, see below)

Page8 – “In our experiments, however, neither rise times (SOM: 5.96 ± 1.22 ms ($n=11$), PV: 5.77 ± 0.99 ms ($n=11$)), nor decay times (SOM: 179 ± 34.5 ms, PV: 199.8 ± 21.6 ms) differed between the two groups (Fig. 5a)” The statistical comparison is missing.

Page9 - “This agonist had no effect on the firing frequency of layerII principal cells (19.8 ± 1.9 Hz control vs 18.4 ± 2.4 Hz J-2156 @300 pA injected current, $n=9$, Fig. 5g,h), but decreased the firing frequency in layerIII-V pyramidal cells (15.1 ± 1.2 Hz control vs 11.1 ± 1.3 Hz J-2156 @150 pA, $n=13$, Fig5. g,h)”. The statistics used here are not appropriate. The differences showed on the figure seems quite big, and I have no doubt that it will remain significant, but my point here is to use proper statistics. They authors use the “Wilcoxon rank sum test” 2 times (comparing SOM+ activation and PV+ activation, and SOM+ activation with SOM+ activation in Sstr4 KO). I believe this test is equivalent to a Mann-Whitney U-test, which is not appropriate when there are more than two groups of independent observations. At least the authors should correct p-values.

Our response: We thank the reviewer for pointing out the weaknesses or even absence of statistics. We completely re-analyzed our data. With the proper analysis (paired T-test for the J-2156 drug experiments for example) we actually found that layerII principal cells also decrease their firing frequency after this drug application, however, the effect was much smaller than in layerIII-V pyramidal cells. We corrected our conclusions accordingly.

3/ All along the manuscript, cells from L3 and L5 have been grouped together. However L3 and L5 in MEC are anatomically distinct, have different cell types or clearly distinct input/output connectivity patterns. Why were they grouped together? If they were because the response to inhibitory cells were similar, or because they are just considered as another group from L2 cells, then the authors should say so, and most importantly, they should justify it. Maybe the amplitude of optogenetically-triggered SOM+ IPSPs or the effect of SOM agonists are different between L3 and L5?

Our response: We have grouped L3 and L5 cells together for several reasons. Both L3I and L5 pyramidal cells are assumed to play a major role in memory formation processes. Indeed, their input/output connectivity is partially different, however, they show one important common feature: the percentage of grid-cells in both layerIII and LayerV is extremely low. Moreover, there is no distinctive electrophysiological feature, which enables us to differentiate between these two layers in in vivo conditions. We have discussed these differences in the current version of the manuscript. We reanalyzed our data and discuss in the manuscript that L3 and L5 cells show similar effects both in PV-ChR2 and SOM-ChR2 animals.

4/ The authors wrote “LayerII pyramidal and stellate cells are the closest located to layerI, where the most somatostatin immunoreactive axonal cloud can be found (Supplementary Fig. 1))” (page 5)

On Supplementary Fig. 1, I don't see the difference of somatostatin expression between layer 1 and 2, neither in a nor b. I don't deny that it exists, but the figure does not show it. Maybe the authors could consider showing a non-overlaid version of a. Adding a simple quantification of the background fluorescent signal along the radial axis and aligned with the figure could help.

The author mention the repartition of SOM fluorescent signal in EC, but not S1 and CA1. I am curious, is there anything in the fluorescent signal repartition that could have predicted the difference of responses between layers in EC and the similarity of the responses in S1 and CA1?

Our response: We have replaced the original Supplementary Fig 1 with a new version where we show the PV and SOM signals in two, non-overlaid black and white images of the same area. Here one can see that SOM immunoreactivity is localised mostly in the cell bodies (from layerII-V), and some of the axons, which are the densest in layerI. On the other hand, SOM immunoreactivity is rather elusive (highly fixation dependent, and only the stronger axons can be visualized with the conventional immunohistochemical methods), therefore, in order to get a better impression of the SOM expressing GABAergic cell location and morphology, we have also included a SOM-Cre driven EGFP expression in the MEC together with a marker, which outlines layerII (WFS1+ layerII pyramidal cells).

Martinotti-cells in the cortex and OLM (oriens-lacunosum moleculare) cells in the hippocampus are well known to target distal dendrites (L1 in cortex and lacunosum moleculare in CA1), however, there are other SOM+ expressing neurons both in the cortex (non-Martinotti cells) and CA1 (bistratified cells) which are targeting less distal dendrites. Because all these cell groups innervate layers where all pyramidal (and stellate) cells have dendrites, there is no obvious repartition indicating the stronger innervation of deeper located pyramidal cells. Our original hypothesis at the beginning of this project was that LayerII cells might have stronger dendritic innervation. We were really surprised to see just the opposite.

5/ The results part should be improved for clarity. The authors should checked that they clearly mention the methods that they used and that the figures are ordered optimally.

Here are listed a few things/suggestions:

The sequence Fig 3 Fig 4 Fig 5, and corresponding text could be revised. Figure 5d contains similar results than figure 3. Maybe Fig 3 and 5 should be grouped together and figure 4 could alternatively come after or as a supplementary.

I noticed that the text describes the results of stimulation of SOM cells before PV cells whereas figure 1 order is the opposite. Reordering the text and the figure panels consistently would make sense. It seems that the authors change the order of the text given that more details about the methods are given in the second paragraph about PV. If the text is kept that way, these method details should be move to the SOM paragraph.

Page 6 “ Here, we utilized the general somatostatin promoter driven Cre mouse line (SOM-Cre) to express ChR2 in all types of SOM cells in both somatosensory cortex and dorsal hippocampus CA1“ It seems to be the same technique as used earlier, but it is not clear how ChR2 is brought here (breeding mouse lines, injection of AAV).

More generally, it is sometimes not clear whether the expression of ChR2 is obtained after breeding or viral injections (both are described in the methods, but what is the approach should be clearly stated in the results and figures when mentioned).

Our response: We substantially rewrote the results section. We have clarified the ChR2 expression method in the S1 and CA1 regions. We reordered figure1, now the results in SOM-ChR2 mice are on the left side and PV-ChR2 animals are on the right side.

Indeed, Fig3 and Fig5 are closely related to each other, and one of the plots in Fig3 is re-plotted in Fig5. In order to focus the reader's attention to the unexpected elongated inhibitory effect, however, we believe that first showing the major difference between SOM-ChR2 and PV-ChR2 is critical. We think that showing the SOM immunoreactivity localized in GABAergic synaptic vesicle expressing boutons is also very important, because most readers are unaware of this phenomenon. Putting figure 3 and 5 into one figure would overcrowd the figure, thus we decided to keep the original arrangement. Moreover, we added action potential examples to figure3 in order to show how we decided about the cells' identity in the silicon probe experiments.

6/ Can the authors briefly justify their use of SST-R4 KO mouse? There are 5 types of SST receptors, I am just wondering why this mouse in particular. Is it present in MEC L3-5? Is it more specific to MEC than other receptors? Easy access to that particular line would also be a legit justification as long as SST-R4 is expressed in all layers of MEC.

Our response: Done.

We have added references and clarification in the results and in the discussion. Unfortunately, there are no specific antibodies against SOM receptors and also no comparative studies have been made in this brain area. SST4 receptor is the most widely expressed out of the 5 types.

7/ The role of somatostatin on the modulation of neuronal activity is only mentioned in the result section, but it should be described in the introduction.

Our response: Done.

We have improved the introduction and added more references on the topic of somatostatin as a neuromodulator.

8/ The main results should be briefly summarized at the beginning of the discussion.

Our response: Done.

We have completely rewritten the discussion, starting with summarizing the results and putting into context with the current literature.

9/ The discussion sometimes reads more like an introduction. There are a lot of references to the literature (which is of course essential), but maybe too much (some parts are actually repeated from the introduction). My point is that it is interesting, but I feel like some of the current paper's results are a bit lost in it, rather than being emphasized.

Our response: Done.

We fully reorganized the discussion and focused more on the findings and the outlooks.

10/ Page 5. "poor space-clamping (intracellular solution contains CsCl, see methods)"
Cs solution limits space clamp, but it does not prevent it at all, especially in distal dendrites (<http://www.nature.com/articles/nn.2137>). Please, remove that statement, and discuss space clamp as a potential but minor limitation of that experiment (I think that it could be done in the methods as a justification for using cs-based internal).

Our response: Done.

We have removed this statement and discussed the limitations.

11/ The effect was monosynaptic on both stellate and pyramidal cells (1.9 ± 0.12 ms and 2 ± 0.17 ms respectively) ; I would use “latency of events” rather than “effect”.

Our response: Done.

Corrected.

12/ The latency of PV+ and SOM+ evoked IPSPs are mentioned for Layer 2 cells but not for 3-5 cells. Also it was not mentioned for CA1 and S1 cells I believe.

Our response: Done

We have added the latency values for layerIII-V, CA1 and S1 as well.

13/ Figure 5c. Even though the indication of the light pulse could be used as a time scale, a proper time scale bar is missing.

Our response: Done.

14/ The authors should briefly describe, in the result section, what protocol they use for light delivery during the behavioral experiment and, ideally, they. It is not clear in the methods for how long the light stimulations (bilateral MEC, 50ms, 5Hz) were delivered (for both Y maze and Morris WM).

Our response: Done.

We have added a description of light delivering protocol to the results section as well.

15/ I thought that the figures were nice and clear.

Just for Figure 5d, 5f, or 6e, I am not a big fan of the bar-graph representation, I think showing all the data points and the median (or mean depending on the statistical test you choose), as in figure 2, would be much nicer. Box/whisker plots are also an alternative. Also in 5h (and 5d), please indicate what the error bars represent.

Our response: Done.

We have replaced the bar graphs with proper plots.

Reviewers' Comments:

Reviewer #1:

Remarks to the Author:

The present manuscript is revised adequately. I appreciate the authors for their collaboration. Detailed morphological analysis was added in this second version of the paper, and it will be of interest to readers working in the field of neuroanatomy. Overall, I think this is an interesting and important paper that would be appropriate for Communications Biology.

Reviewer #2:

Remarks to the Author:

For the most part my previous comments have been adequately addressed.

There are some issues that require attention.

1. I could not find in the Methods any clear statement for how group sizes were determined or whether the experimenters were blind to the group or the manipulation. The group size statement should clearly indicate whether power analyses were carried out.
2. The author's did not address my question about whether Water maze and T maze experiments used the same or different animals. This should be clearly stated.
3. The Discussion should highlight that the experimental design did not allow confirmation that, in the behavioural experiments, ChR2 was similarly effective between groups. The experiments presented are a nice starting point, but this is a major caveat in their interpretation.

Reviewer #3:

Remarks to the Author:

The authors addressed most of my concerns and the manuscript has been greatly improved.

I still have a few points mostly concerning the clarity of the paper. I think that will not require too much work.

1/

My first main concern is related to one I had with the first version, about grouping layer III and layer V MEC cells. The authors' justification for grouping the cells together makes sense to me, and I have no problem with that. In this revised version, the authors even provide an additional reason to group them together by showing that the synaptic responses to optogenetics activation of interneurons are similar for layer III and layer V cells. The problem is that they do the comparison in the text, but not in the figure.

It is an important result that should be highlighted in figure 1, also to make text and figure consistent. Besides, the text corresponding to this part should be revised (see my second point).

2/

My second main concern is the readability of the manuscript, mostly of the results section. It was significantly changed during the revision, it was improved, but it still requires significant polishing. For example, some sentences are quite long and "heavy", the choice of some words could be improved, and there are numerous language mistakes. I have noted a few examples (see the few points below) so the authors can understand what I am referring to.

Besides, I recommend asking a native English speaker for feedback before resubmission.

(i)

Line 138:

"For this, first we tested the postsynaptic targets of mCherry-tagged SOM-ChR2+ synaptic boutons in all investigated layers with electron-microscopy."

"tested"? Nothing was "tested" here. "checked", "described" maybe

(ii)

The word "effect" was used many times in the text whereas it could be replaced by more specific terms. For example, line 148, you could replace "effect" by "delay", or line 155, you could replace "effect" by "evoked PSP amplitudes". That would make the text clearer.

(iii)

Line 179: The conclusion here is not very sharp.

"Therefore, we concluded that in contrary to previous immunohistochemical predictions, deep layer (layerIII-V) pyramidal cells receive strong PV+ innervation, thus PV+ GABAergic cells have an overall strong GABAergic effect on all principal (pyramidal and stellate) cell types in layerII-V of the MEC".

"in contrary to" is not proper English.

"GABAergic effect" is very vague and could refer to many things. It could be hyperpolarization, inhibitory current amplitude, shunting, inhibition of firing...

(iv)

The authors now report the number of recordings/cells as well as the number of animals, as asked by another reviewer. They write "(n = 6, N = 5)". I would suggest to be clear about what "n" and "N" are referring to; e.g. "(n = 6 cells in N = 5 animals)" or "(n = 4 WFS1+ cells in N = 3 mice)"

(v)

Generally, if there is a p-value somewhere, the test should be written next to it. Also, statistical tests should be used only if it makes sense, i.e. if a comparison is made.

e.g. Line 149: one p values is reported; what test is it? Also, there was no comparison in the sentence just before so I don't understand why there is a p-value here.

(vi)

Line 154:

"By contrast, layerIII (n=15, N=12) and layerV (n=8, N=6) pyramidal cells responded with a magnitude higher amplitude to the same duration and intensity of whole field illumination (LIII: 13.58 v155 ±2.02 mV, LV: 16.6 ±2.09 mV, p=0.31 Fig. 1h-j)."

The sense is kind of clear but the English isn't.

Also, what is the p-value referring to? What test? And there was no prior mention of any comparison... (I guess that it is to show that layer III and layer V cells receive similar postsynaptic event in response to whole field illumination, but I should not have to guess)

3/

In the methods, line 556.

Please report how the data was tested for normality?

4/

Line 162: "the apical dendrites of LayerVI pyramidal cells do not enter layerI-V"

The authors refer to the ref 24. I am not sure if it is the most appropriate reference for that exact statement. <http://doi.wiley.com/10.1002/hipo.20993>, from the same authors, might be better (but maybe I am wrong).

5/

Line 258: where is sup fig 3? Do you mean sup fig 1?

6/

The title is quite broad and could mean many things. It should sound more functional.

Good luck with the revision

Response to referees

We thank the reviewers constructive comments on the latest version of our manuscript.

REVIEWERS' COMMENTS:

Reviewer #1 (Remarks to the Author):

The present manuscript is revised adequately. I appreciate the authors for their collaboration. Detailed morphological analysis was added in this second version of the paper, and it will be of interest to readers working in the field of neuroanatomy. Overall, I think this is an interesting and important paper that would be appropriate for Communications Biology.

Our response: We appreciate your constructive critique and comments on the previous version of the manuscript.

Reviewer #2 (Remarks to the Author):

For the most part my previous comments have been adequately addressed.

There are some issues that require attention.

1. I could not find in the Methods any clear statement for how group sizes were determined or whether the experimenters were blind to the group or the manipulation. The group size statement should clearly indicate whether power analyses were carried out.

Our response: We made a clear statement in the current version about the fact that we have not carried out power analysis prior the experiments, but our sample sizes are similar to those reported in previous publications in the field.

2. The author's did not address my question about whether Water maze and T maze experiments used the same or different animals. This should be clearly stated.

Our response: We describe in the current version of the manuscript in the Methods, on which animals were used for both experiments.

3. The Discussion should highlight that the experimental design did not allow confirmation that, in the behavioural experiments, ChR2 was similarly effective between groups. The experiments presented are a nice starting point, but this is a major caveat in their interpretation.

Our response: We added a statement to the discussion about this potential caveat.

Reviewer #3 (Remarks to the Author):

The authors addressed most of my concerns and the manuscript has been greatly improved.

I still have a few points mostly concerning the clarity of the paper. I think that will not require too much work.

1/

My first main concern is related to one I had with the first version, about grouping layerIII and layerV MEC cells. The authors' justification for grouping the cells together makes sense to me, and I have no problem with that. In this revised version, the authors even provide an additional reason to group them together by showing that the synaptic responses to optogenetics activation of interneurons are similar for layer III and layer V cells. The problem is that they do the comparison in the text, but not in the figure. It is an important result that should be highlighted in figure 1, also to make text and figure consistent. Besides, the text corresponding to this part should be revised (see my second point).

Our response: We have added images of recorded layerV pyramidal cells and also their responses to light stimulation. We have changed Fig 1 accordingly.

2/

My second main concern is the readability of the manuscript, mostly of the results section. It was significantly changed during the revision, it was improved, but it still requires significant polishing. For example, some sentences are quite long and "heavy", the choice of some words could be improved, and there are numerous language mistakes. I have noted a few examples (see the few points below) so the authors can understand what I am referring to.

Besides, I recommend asking a native English speaker for feedback before resubmission.

Our response: we thank the Reviewer for drawing our attention to the language mistakes. We have incorporated the Reviewer's suggestions, and have consulted native English speakers to improve readability and clarity of the text.

(i)

Line 138:

"For this, first we tested the postsynaptic targets of mCherry-tagged SOM-ChR2+ synaptic boutons in all investigated layers with electron-microscopy."

"tested"? Nothing was "tested" here. "checked", "described" maybe

Our response: we replaced “tested” by “checked”.

(ii)

The word “effect” was used many times in the text whereas it could be replaced by more specific terms. For example, line 148, you could replace “effect” by “delay”, or line 155, you could replace “effect” by “evoked PSP amplitudes”. That would make the text clearer.

Our response: we replaces the word “effect” by appropriate terms.

(iii)

Line 179: The conclusion here is not very sharp.

“Therefore, we concluded that in contrary to previous immunohistochemical predictions, deep layer (layerIII-V) pyramidal cells receive strong PV+ innervation, thus PV+ GABAergic cells have an overall strong GABAergic effect on all principal (pyramidal and stellate) cell types in layerII-V of the MEC”.

“in contrary to” is not proper English.

“GABAergic effect” is very vague and could refer to many things. It could be hyperpolarization, inhibitory current amplitude, shunting, inhibition of firing...

Our response: We have modified this sentence accordingly.

(iv)

The authors now report the number of recordings/cells as well as the number of animals, as asked by another reviewer. They write “(n = 6, N = 5)”. I would suggest to be clear about what “n” and “N” are referring to; e.g. “(n = 6 cells in N = 5 animals)” or “(n = 4 WFS1+ cells in N = 3 mice)”

Our response: we specify in the current version of the manuscript in the methods what N and n stands for.

(v)

Generally, if there is a p-value somewhere, the test should be written next to it. Also, statistical tests should be used only if it makes sense, i.e. if a comparison is made.

e.g. Line 149: one p values is reported; what test is it? Also, there was no comparison in the sentence just before so I don’t understand why there is a p-value here.

Our response: we have removed from the current version of the manuscript the unnecessary statistics and p values.

(vi)

Line 154:

“By contrast, layerIII (n=15, N=12) and layerV (n=8, N=6) pyramidal cells responded with a magnitude higher amplitude to the same duration and intensity of whole field illumination (LIII: 13.58 ± 2.02 mV, LV: 16.6 ± 2.09 mV, $p=0.31$ Fig. 1h-j).”

The sense is kind of clear but the English isn’t.

Also, what is the p-value referring to? What test? And there was no prior mention of any comparison... (I guess that it is to show that layer III and layer V cells receive similar postsynaptic event in response to whole field illumination, but I should not have to guess)

Our response: we have modified this sentence and removed unnecessary comparisons (and p values).

3/

In the methods, line 556.

Please report how the data was tested for normality?

Our response: we specify in the current version the type of test for data normality.

4/

Line 162: "the apical dendrites of LayerVI pyramidal cells do not enter layerI-V"

The authors refer to the ref 24. I am not sure if it is the most appropriate reference for that exact statement. <http://doi.wiley.com/10.1002/hipo.20993>, from the same authors, might be better (but maybe I am wrong).

Our response: We agree with the suggested replacement of the reference.

5/

Line 258: where is sup fig 3? Do you mean sup fig 1?

Our response: We have corrected this to refer to the correct figure number (Fig 4).

6/

The title is quite broad and could mean many things. It should sound more functional.

Our response: we have changed the title of the manuscript to be more specific about the results of the study.